# Differences in cortical activation patterns during action observation, action execution, and interpersonal synchrony between children with or without autism spectrum disorder (ASD): An fNIRS pilot study

Wan-Chun Su[1,2], McKenzie Culotta[1,2], Jessica Mueller[3], Daisuke Tsuzuki[4], Kevin Pelphrey[5], Anjana Bhat[1,2,6¤]*

1 Department of Physical Therapy, University of Delaware, Newark, Delaware, United States of America, 2 Biomechanics & Movement Science Program, University of Delaware, Newark, Delaware, United States of America, 3 Department of Behavioral Health, Swank Autism Center, A. I. du Pont Nemours Hospital for Children, Wilmington, Delaware, United States of America, 4 Department of Language Sciences, Tokyo Metropolitan University, Tokyo, Japan, 5 Department of Neurology, University of Virginia, Charlottesville, Virginia, United States of America, 6 Department of Psychological & Brain Sciences, University of Delaware, Newark, Delaware, United States of America

¤ Current address: University of Delaware, Newark, Delaware, United States of America.
* abhat@udel.edu

## Abstract

Engaging in socially embedded actions such as imitation and interpersonal synchrony facilitates relationships with peers and caregivers. Imitation and interpersonal synchrony impairments of children with Autism Spectrum Disorder (ASD) might contribute to their difficulties in connecting and learning from others. Previous fMRI studies investigated cortical activation in children with ASD during finger/hand movement imitation; however, we do not know whether these findings generalize to naturalistic face-to-face imitation/interpersonal synchrony tasks. Using functional near infrared spectroscopy (fNIRS), the current study assessed the cortical activation of children with and without ASD during a face-to-face interpersonal synchrony task. Fourteen children with ASD and 17 typically developing (TD) children completed three conditions: a) *Watch*—observed an adult clean up blocks; b) *Do*—cleaned up the blocks on their own; and c) *Together*—synchronized their block clean up actions to that of an adult. Children with ASD showed lower spatial and temporal synchrony accuracies but intact motor accuracy during the Together/interpersonal synchrony condition. In terms of cortical activation, children with ASD had hypoactivation in the middle and inferior frontal gyri (MIFG) as well as middle and superior temporal gyri (MSTG) while showing hyperactivation in the inferior parietal cortices/lobule (IPL) compared to the TD children. During the Together condition, the TD children showed bilaterally symmetrical activation whereas children with ASD showed more left-lateralized activation over MIFG and right-lateralized activation over MSTG. Additionally, using ADOS scores, in children with ASD greater social affect impairment was associated with lower activation in the left MIFG and more repetitive behavior impairment was associated with greater activation over bilateral

**Data Availability Statement:** All relevant data are within the manuscript and its Supporting Information files.

**Funding:** This work was supported by the National Institutes of Health through a shared instrumentation grant awarded to the University of Delaware (Grant #: 1S10OD021534-01, PI: Bhat) and pilot award funding through an Institutional Development Award (IDeA) from the National Institute of General Medical Sciences of the National Institutes of Health (U54-GM104941, PI: Binder-Macleod; P20 GM103446, PI: Stanhope). AB also thanks the Dana foundation for their support of this fNIRS-based research through a Clinical Neuroscience Award. The funders had no role in study design, data collection and analysis, decision to publish, or preparation of the manuscript.

**Competing interests:** No authors have any competing interests.

MSTG. In children with ASD better communication performance on the VABS was associated with greater MIFG and/or MSTG activation. We identified objective neural biomarkers that could be utilized as outcome predictors or treatment response indicators in future intervention studies.

## Introduction

Autism Spectrum Disorder (ASD) is a neurodevelopmental disorder characterized by impairments in social communication as well as restricted and repetitive behaviors/interests [1]. Children with ASD show impairments in verbal and non-verbal communication skills such as language delays and reduced social gaze/joint attention (i.e., difficulty shifting attention between objects and people) [2–4]. Apart from the diagnostic social impairments, children with ASD have a variety of motor difficulties, including impaired upper/lower limb coordination and postural control [5–7]. They also have difficulties engaging in socially embedded movements, such as imitation [8, 9] and interpersonal synchrony [10–12]. In this study, we specifically focused on interpersonal synchrony performance given its importance in facilitating relationships with peers and caregivers [13, 14]. Additionally, we also examined the underlying neural impairments associated with interpersonal synchrony performance in children with and without ASD. While there is neuroscientific literature on imitation performance that could be extended to interpersonal synchrony behaviors, it is still limited to imitation of finger motions and does not include naturalistic arm movements and face-to-face interactions [15, 16]. Hence, the present study aimed to compare interpersonal synchrony performance as well as underlying cortical activation patterns during a naturalistic reach and clean up task in children with and without ASD.

Imitation and interpersonal synchrony share similar perceptuo-motor integration processes [14]. For both imitation and interpersonal synchrony, one needs to perceive the cues from the environment and their partner, then anticipatorily control and reactively adjust their actions to match the actions of their partner [17, 18]. In short, the components of observation/perception and motor execution (motor planning and control) are common to both behaviors. Both behaviors require children to copy the spatial form of actions performed by the partner (i.e., spatial accuracy); however, they differ in temporal aspects. While imitation involves copying of discrete limb and body movements, interpersonal synchrony requires two individuals to move synchronously over time with actions of both partners being time-constrained (i.e., related to one another in time) [14, 19]. Apart from spatial accuracy, interpersonal synchrony also involves analyses of temporal accuracy (i.e., are both individuals moving similarly in speed and phasing). The challenge of interpersonal synchrony is greater than that of imitation due to the continuous nature of actions performed and the motor control challenges associated with it. For example, musicians synchronize their actions on instruments to produce beautiful orchestral sounds or two people work together to lift large and heavy objects simultaneously [20, 21]. Therefore, in the current study, we not only measured the spatial accuracy but also the temporal accuracy during interpersonal synchrony behaviors in children with and without ASD. Moreover, we compared the behaviors and associated cortical activation between interpersonal synchrony and its component behaviors (i.e., action observation and execution). Our past findings in adults and typically developing children show that the neural complexity of imitation/interpersonal synchrony arises from the motor component and not the perceptual component [22, 23].

As mentioned earlier, while there is limited literature on brain activation patterns associated with interpersonal synchrony behaviors, there is substantial evidence describing the

neural mechanisms associated with imitation behaviors. Both imitation and interpersonal synchrony share similar basic processes of monitoring/perceiving, anticipating/planning, and executing actions; hence, the underlying neural substrates are most likely similar [14]. fMRI studies confirm widespread cortical activation during imitation and its component behaviors of observation and execution over bilateral frontal, parietal, and temporo-occipital regions [24–26]. Certain cortical regions are said to be consistently active during action observation, action execution, and imitation, forming an important imitation network [27, 28] including the i) the Inferior Frontal Gyrus (IFG) and ventral Premotor Cortex (vPMC) of the frontal lobe, ii) the Superior Temporal Sulcus (STS), Superior and Middle Temporal Gyri (STG and MTG) of the temporal lobe, and iii) the Inferior Parietal Lobule (IPL), including the intraparietal sulcus of the parietal lobe, the Supramarginal Gyrus (SMG) and the Angular Gyrus (AG). Additional brain regions activated during imitation behaviors may include other visual, social, and motor regions important for visual/social perception, working memory, motor planning, and action execution including dorsolateral prefrontal cortices, premotor cortices, primary and supplementary/pre-supplementary motor cortices, cingulate/insular cortices, cuneus/precuneus as well as subcortical structures such as the cerebellum and putamen [29–34].

During imitation/interpersonal synchrony, each of the aforementioned regions are said to play different roles. The STS is more active during observation of biological than non-biological motions and is said to provide a visual description of observed actions as well as compare the observed and planned actions [35, 36]. Both IFG and IPL are active during observation, execution, and imitation of goal-directed object manipulation [30, 37]. The IPL may contribute to planning the kinematics of a goal-directed action; while IFG is said to play an important role in goal understanding [24, 30, 38]. The original studies reported various cortical regions that consistently activated during action observation, execution, and imitation, without highlighting the differences in activation levels across conditions [39]. While early studies suggested greatest cortical activation during action imitation followed by action execution and lastly, action observation [40], more variable levels of activation have been reported across different regions and tasks [36, 41, 42]. The STS region was consistently found to be more active during imitation compared to action execution and observation across different tasks, including actions involving simple finger movements [41], pantomimed actions on objects [36, 42], and communicative gestures [42]. The IPL was more active during conditions involving movements (execution and imitation) compared to action observation of simple finger movements [43, 44], pantomimed actions on objects [36, 42], and communicative gestures [42]. The results were inconsistent in the IFG region depending on the nature of the task. During an object grasping task there was higher IFG activation during action imitation than action execution and observation [45]. However, during pantomimed actions on objects and communicative gesture tasks there was greater IFG activation during action imitation and execution than action observation [42]. These inconsistent results might be due to the variable roles played by these regions across different tasks/contexts. The STS region is important for matching observed movements with one's own movement, and therefore, is more activated during imitation than other conditions; whereas the IPL region encodes the kinematic quality of movements, and therefore would be more activated during movement conditions. The IFG region encodes the goal of the imitated action, and therefore, differs in activation across tasks with different goals.

Some have suggested that the neural circuitry associated with imitation behaviors may be an evolutionary precursor to the language system, and therefore, is left-hemisphere dominant [46, 47]. The majority of the fMRI studies investigating brain activation during imitation only involve the right hand, limiting the ability to compare activation between hemispheres [41, 45, 48]. Others have found that even during unilateral finger movements, there is bilateral

activation during action observation and imitation highlighting the bilateral nature of the imitation network [27]. The bilateral pattern was further confirmed by a study comparing imitation-related activation when using left or right hands [40, 49]. During the imitation condition, Aziz-Zadeh and colleagues found bilateral activation in the IFG and IPL with more ipsilateral than contralateral activation depending on which hand moved [40]; however, there was always greater right STS activation regardless of the moving hand. Similar right lateralization has been reported in the STS region when processing biological motions [35]. In contrast, during gestural imitation, Mühlau et al. found more bilateral IFG and STS activation and more left-lateralized activation in the IPL region [49].

In spite of the robust neural mechanisms related to imitation, there are few studies on the neural correlates of interpersonal synchrony behaviors due to the difficulties in displaying and performing natural synchronous actions between partners from within the MRI scanner. fMRI researchers have adapted their tasks to perceived synchrony paradigms with a virtual partner during finger tapping motions [50, 51]. When a participant better synchronized with a virtual partner, Cacioppo et al. found greater activation over the IPL, parahippocampal gyrus, and ventromedial prefrontal cortex while Fairhurst et al. found greater IFG activation [50, 51]. Using functional near-infrared spectroscopy (fNIRS), it is possible to conduct face-to-face interactions with a human partner while performing arm movements. A study involving healthy adults showed higher activation over the IPL region as they synchronized their actions with a partner during a table setting task requiring transporting of tableware jointly with a partner compared to solo actions involving transport of tableware [52]. Using a block clean up task, we found greater right IPL and right IFG activation in healthy adults during interpersonal synchrony compared to a solo, action execution condition [22]. Taken together, both fMRI and fNIRS studies have provided contrasting findings on conditional and hemispheric differences in cortical activation with more studies confirming a pattern of bilateral activation during imitative/interpersonal synchrony behaviors.

Children with ASD have impairments in synchronizing their actions with that of a social partner [10–12]. To be more specific, children with ASD spend less time synchronizing their actions with their social partners during intentional synchrony of marching and clapping actions (i.e., synchronizing arm and leg actions to the marching and clapping actions of a partner) [11] as well as unintentional synchrony when spontaneously rocking in a chair (i.e., unintentionally synchronizing rocking motions to that of the caregiver while reading a story book) [12]. They also showed reduced quality of interpersonal synchrony during pendulum swaying synchrony tasks (i.e., swaying of the pendulum antero-posteriorly while synchronizing with the tester), with more variable and lagged movements [10]. Children with ASD might have poor social attention [53] as well as poor visuo-motor coordination skills [54] that affects their ability to continuously match their actions with their partners within social contexts. Although not consistent, fMRI studies have reported that children with ASD have atypical cortical activation during imitation tasks. When imitating hand gestures, children with ASD showed reduced activation in the right angular gyrus, precentral gyrus, and left middle cingulate gyrus [15]. During finger movement imitation, children with ASD had reduced activation over right fusiform cortex, right middle occipital gyrus, left IPL, right lingual gyrus, right middle temporal gyrus [55], as well as the cerebellum [16]. A recent meta-analysis involving imitation tasks found that individuals with ASD had increased IPL activation and altered activation over the occipital, dorsolateral prefrontal, and cingulate cortices, as well as the insula, compared to control participants [56]. An EEG study investigating brain activation patterns during interpersonal synchrony in children with ASD [57] found that during the baseline period before starting the auditory finger tapping synchrony task (i.e., task of synchronizing finger tapping movements with a partner/computer), children with ASD showed increased theta activity

associated with midline prefrontal cortex activation with no differences between groups during the synchronized tapping period itself [57]. The prefrontal cortex contributes to executive functions and motor planning suggesting that the children with ASD might engage in greater motor planning/executive functioning as they planned for synchronized actions.

Although fMRI provides an accurate functional analysis of the whole brain with good spatial resolution, it requires participants to lie still in a noisy scanner. Therefore, the previous fMRI studies were constrained to the imitation of simple hand movements [36, 41, 42] or perceived interpersonal synchrony of a partner [50, 51] without involving naturalistic social interactions. Furthermore, although there are a growing number of studies utilizing fMRI in the children with ASD, the fMRI testing environment is still challenging for children with ASD, leading to greater anxiety and poor task compliance due to the loud noise and narrow space of the scanner bore [58]. In contrast, fNIRS is a non-invasive optical neuroimaging tool that has been applied to various motor skills such as walking [59], dancing [60], as well as arm movements [52, 61]. It also allows participants to engage in face-to-face interactions while staying upright and tolerates movement artifacts. Using fNIRS, in the current study, we recorded cortical activation during upright, face-to-face interactions between tester-child pairs.

Overall, the broad goal of the current study was to compare the behavioral differences and associated cortical activation between children with and without ASD during action observation, execution, and interpersonal synchrony in a naturalistic, reach and clean-up task. We hope to identify neural biomarkers of interpersonal synchrony impairments in children with ASD that could be used as objective measures to examine intervention outcomes in future studies. In terms of the conditional differences, we hypothesized that both TD and children with ASD would have higher activation during interpersonal synchrony compared to action execution; and higher activation during action execution compared to action observation. For hemispheric differences, we expected TD children to show more bilateral activation while children with ASD would show more asymmetrical activation during interpersonal synchrony. In terms of group differences, we hypothesized that children with ASD would have lower activation in MIFG and MSTG regions and increased activation over the IPL regions compared to the TD children. Given the variable nature of ASD symptoms, we expected cortical activation in children with ASD to be associated with their ASD severity and level of adaptive functioning.

## Materials and methods

### Participants

Fourteen children with Autism Spectrum Disorder (ASD) (mean age ± SE: 11.29 ± 0.93, 9 males and 5 females), and 17 age-and-sex matched Typically Developing Children (TD) (mean age ± SE: 10.82 ± 0.69, 11 males and 6 females, no group differences for age and sex, see Table 1 for more details) were recruited in this study. Participants were recruited through online announcements, phone calls and fliers distributed to various local schools, community centers, local autism services, and ASD advocacy groups. We completed screening interviews with all children to obtain their demographic information including age, sex, ethnicity, socioeconomic status, and handedness as well as to confirm their eligibility for participation (Table 1). The inclusion criteria for children with ASD were: a) holding an ASD diagnosis offered by a professional (i.e., neurologist, psychologist, psychiatrist) and confirmed through medical or school records, b) having the ability to follow one-step instructions such as "move like this", c) a lack of significant behavioral issues, e.g., difficulty wearing a cap and inability to remain seated for ~ 30 minutes. TD children were age and sex matched to children with ASD. The exclusion criteria for TD children were: a) having any neurological or developmental

**Table 1. Demographic, SES-Child, handedness, ADOS, VABS-II, and IQ scores of ASD and TD children.**

| Characteristics | ASD (n = 14) Mean ± SE | TD (n = 17) Mean ± SE |
|---|---|---|
| Age | 11.29 ± 0.93 | 10.82 ± 0.69 |
| Sex | 9M; 5 F | 11 M; 6 F |
| Ethnicity | 10 C; 2A;1BC;1AC | 13 C; 1A; 1AI; 2AC |
| SES-Child | 68.86 ± 4.49 | 69.71 ± 4.43 |
| Coren's Handedness Score | 13 R, 1 L 33.57 ± 1.60 | 15R, 2L 33.41 ± 1.78 |
| SCQ | 25.64 ± 9.31 | - |
| ADOS | 18.17 ± 1.86 | |
| Social affect | 13.69 ± 1.41 | |
| Repetitive Behavior | 5.23 ± 0.61 | |
| VABS (SS) | 70.57 ± 3.39* | 110.29 ± 2.92 |
| Communication (SS) | 72.79 ± 3.46* | 109.82 ± 2.88 |
| Daily living (SS) | 75.86 ± 4.01* | 110.41 ± 3.08 |
| Socialization (SS) | 67.93 ± 4.06* | 106.53 ± 3.18 |
| Stanford-Binet IQ | | |
| Full scale IQ | 79.57 ± 6.77* | 114.18 ± 1.71 |
| Verbal IQ | 83.62 ± 7.54* | 114.59 ± 2.25 |
| Non-verbal IQ | 69.10 ± 6.69* | 114.53 ± 1.74 |

SES-Child = Hollingshead Four-Factor Index of Socioeconomic Status; SCQ = Social Communication Questionnaire; ADOS = Autism Diagnostic Observation Schedule - 2nd Edition; VABS = Vineland Adaptive Behavior Scale - 2nd Edition; SS = Standard Score; IQ = Intelligence Quotient; M = Male, F = Female; R = right, L = left; C = Caucasian, A = Asian, AI = American Indian; BC = Black-Caucasian; AC = Asian-Caucasian *indicates significant differences between ASD and TD groups.

diagnoses/delays, preterm birth, or significant birth history, b) taking medications with neural or psychotropic effects, c) having a history of seizures, d) having uncorrected vision or hearing impairments and d) having a family history of ASD.

We confirmed the diagnosis of ASD through medical/neuropsychological/school records and/or the presence of a social communication delay using the Social Communication Questionnaire [62] (SCQ, averaged score ± SE: 25.64 ± 9.31). In addition, a clinical psychologist (i.e., 3rd author) independently confirmed the diagnosis of ASD using the Autism Diagnostic Observation Schedule– 2nd edition [63] (ADOS, average ADOS score ± SE = 18.17 ± 1.86). She also assessed the level of intelligence in children with and without ASD using the Stanford-Binet IQ test [64] (Full scale IQ ± SE: ASD: 79.57 ± 6.77; TD:114.18±1.71, $p < 0.001$) (Table 1). In addition, the Hollingshead Four-Factor Index of Socioeconomic Status [65] (SES-Child) was used to estimate the socioeconomic status (averaged score ± SE: ASD: 68.86 ± 4.49, TD: 69.71 ± 4.43, $p > 0.05$), while the Coren's handedness survey was used to determine their handedness [66] (average handedness score ± SE: ASD: 33.57 ± 1.60, TD: 33.41 ± 1.78, $p > 0.05$). Thirteen children with ASD were strongly right-handed, with one showing moderate left-handedness. Fifteen of the TD children were found to be strongly right-handed, while two children showed moderate left-handedness. Note that all subjects completed the task using their right hand. The activation patterns of the three left-handed children were similar to the group results; hence, their data have been retained. The parents of the participating children also completed Vineland Adaptive Behavioral Scales-2nd edition questionnaire [67] (VABS) to provide a measure of socialization (averaged score (%) ± SE: ASD: 67.93 ± 4.06, TD: 106.53 ± 3.18, $p < 0.001$), daily living skills (averaged score ± SE: ASD: 75.86 ± 4.01, TD: 110.41 ± 3.08, $p < 0.001$), communication (averaged score ± SE: ASD: 72.79 ± 3.46, TD: 109.82 ± 2.88, $p < 0.001$) as well as overall adaptive functioning (averaged total score ± SE: ASD: 70.57 ± 3.39, TD: 110.29 ± 2.92, $p < 0.001$) of their children (Table 1). Parents of

participants completed consent forms and the participants completed assent forms before participating in this study. These forms were approved by the University of Delaware Institutional Review Board (UD IRB, Study Approval #: 930721–12).

## Experimental procedures

During the experiment, the children were seated at a table face-to-face with an adult tester. Eight blocks with different shapes and colors were arranged along a circle in front of both, the child and the tester (Fig 1A). Children were asked to clean up the blocks into a container placed on the right-side using their right hand. All children completed the three conditions (Watch, Do, and Together) for multiple trials that occurred in a random order (see trial order in Fig 1B). During the Watch condition, the child observed the tester pick up the blocks in a sequential manner and put them into a container. To ensure that the children paid attention during the Watch trials, before beginning the trial, we instructed them to focus on the pattern of clean up. Instructions were, "Watch me carefully as I clean up the blocks." After a Watch trial was completed, they were asked, "Which block did I pick up first? Or which block did I pick up last? Or how did I clean up the blocks, etc." For the Do condition, the participants cleaned up all the blocks in a sequence of their choice using the instruction "You clean up the blocks on your own." In the Together condition, the tester led the block clean up in a random order while asking the child to pick up the corresponding block placed in front of them using the instruction, "Copy me, let's clean up the blocks together." No questions were asked to the child after completing the Do and Together conditions. The participants were asked to use their right hands, while the tester used her left hand. The children completed a total of 18 trials (6 trials per condition that were randomized across the entire session (Fig 1B). The stimulation period comprised of the time the children took to complete the clean-up task (duration (sec.) $\pm$ SE during Watch: ASD: 11.5 $\pm$ 0.6, TD: 10.6 $\pm$ 0.2, $p > 0.05$; Do: ASD: 11.9 $\pm$ 0.9, TD: 10.3 $\pm$ 0.4, $p > 0.05$; Together: ASD: 15.5 $\pm$ 0.9, TD: 13.6 $\pm$ 0.6, $p < 0.05$). A 10-second pre-stimulation and a 16-second post-stimulation period were included to account for any baseline drifts in the fNIRS signal and to allow the hemodynamic response to return to baseline before starting the next trial. During baseline periods, the participants were asked to focus on a crosshair and remain as still as possible.

## Data collection

The hemodynamic changes over the regions of interest (ROI) were recorded using the Hitachi ETG-4000 system (Hitachi Medical Systems, Tokyo, Japan), with a sampling rate of 10Hz.

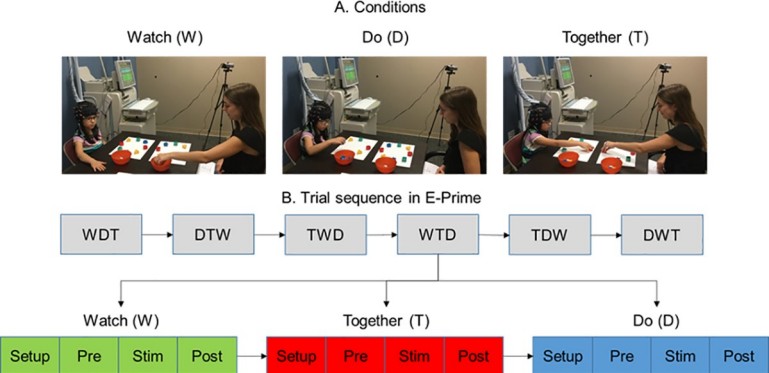

**Fig 1.** Experimental setup (A) and task sequence (B). Written permission for publication of participant pictures has been taken.

Two 3×3 probe sets, consisting of five infrared emitters and four receivers, were positioned over the bilateral frontoparietal and temporal regions. The middle column of the probe set was aligned with the tragus of the ear and the lowermost row of the optode set was aligned with the T3 position of the International 10–20 system [68] (Fig 2A and 2B). As shown in Fig 2A and 2B, the emitters (red) and receivers (blue) were placed in an alternating fashion and each emitter-receiver pair was placed 3 cm apart from each other. The emitters emit two wavelengths of infrared light (695 and 830mm) through the skull creating a banana-shaped arc that reaches the cortical area approximately below the midpoint of the two probes. The midpoint of each emitter-receiver pair forms an fNIRS channel; there are 24 channels in total, 12 on the left side and 12 on the right side, see Fig 1C and 1D). The attenuation of infrared light was used to calculate the changes in concentrations of oxygenated ($HbO_2$) and deoxygenated hemoglobin (HHb) chromophores per channel using the Modified Beer-Lambert Law. Based on past findings, an increase in $HbO_2$ concentration and a decrease in HHb concentration is expected with increased brain activation below a certain channel [69]. E-Prime presentation software (version 2.0) was used to trigger the Hitachi fNIRS system. The entire session was videotaped using a camcorder that was synchronized with the Hitachi fNIRS system.

## Spatial registration approach

For each session, the 3D locations of the standard cranial landmarks (nasion, inion, right and left ears) as well as 3D locations of each probe in the fNIRS probe set were recorded w.r.t. a reference coordinate system using the ETG-4000 system. The anchor-based spatial registration

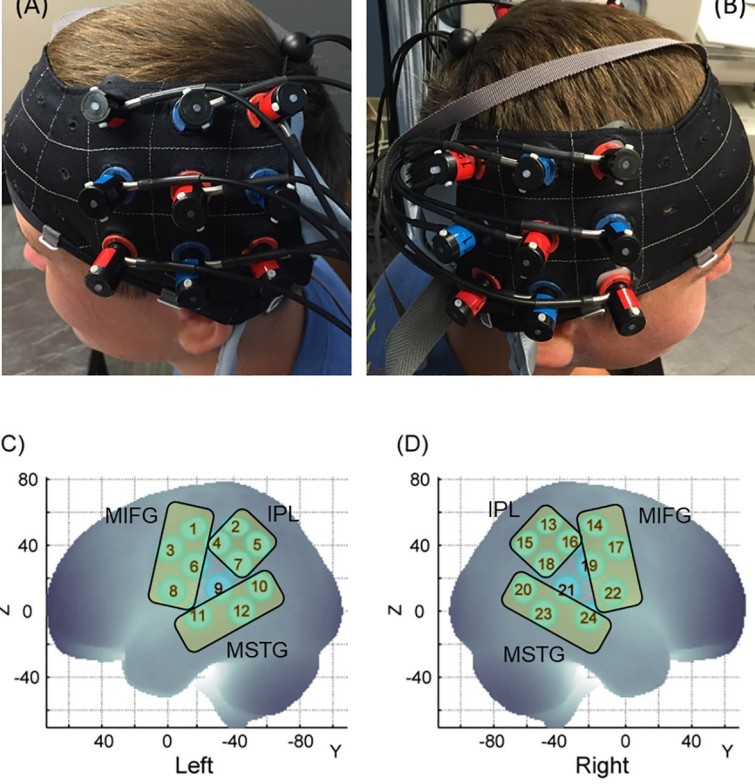

**Fig 2.** Probe placement (A, B) and spatial registration output (C, D). Written permission for publication of participant pictures has been taken.

method developed by Tsuzuki et al. (2012) was used to transform the 3D spatial location of each channel to the Montreal Neurological Institute (MNI)'s coordinate system [70]. The structural information from an anatomical database of 17 adults [71] was then used to provide estimates of channel positions within a standardized 3D brain atlas [70]. The estimated channel locations were anatomically labeled using the LONI Probabilistic Brain Atlas (LPBA) based on MRI scans of 40 healthy adults [72]. Based on the regions covered by our channels, we assigned the 24 channels to three regions of interest (ROI) on each side: i) the frontal region comprised the Middle and Inferior Frontal Gyri (MIFG) and included channels over the inferior/middle frontal gyri, and pre-central gyrus (i.e., channels 1, 3, 6, 8 on the left and channels 14, 17, 19, 22 on the right, see Fig 2C and 2D), ii) the parietal region or Inferior Parietal Lobule (IPL) included channels over the supramarginal gyrus, angular gyrus, and postcentral gyrus (i.e., channels 2, 4, 5, 7 on the left and channels 13, 15, 16, 18 on the right, see Fig 2C and 2D) and iii) the temporal region comprised the Middle and Superior Temporal gyri (MSTG) or the superior temporal sulcus and included channels over the middle and superior temporal gyri (MTG and STG, i.e., channels 10, 11, 12 on the left and channels 20, 23, 24 on the right, see Fig 2C and 2D). As shown in the supplementary materials' S1 Table, channel 21 could not be assigned to any ROI. To avoid inconsistency within the averaged activation data, channel 21, as well as its homologue from the left side (i.e., channel 9) were excluded. In this way, we were able to assign 22 out of the 24 channels to one of the aforementioned ROIs in both groups (details in S1 Table under supplementary materials).

Apart from assessing regional differences using averaged channels, we also conducted channel-specific regional comparisons by using a single representative channel to reconfirm our results. Note that the representative channels for each ROI and hemisphere are bolded in the S1 Table. We acknowledge that fNIRS was unable to perfectly isolate ROIs to a single channel in all cases; however, we were able to isolate 8 out of 10 channels to individual ROIs with a single channel covering 62–100% of the assigned ROI—MFG, IFG, STG, MTG, or IPL. The results of channel-specific analyses were similar to that of averaged channel analyses (see S2 Table under supplementary materials for statistical details).

## Data processing

Customized Matlab programs that incorporated Matlab functions from open-source software such as Hitachi PoTATo [73] and Homer-2 [74] were used to analyze the data output from the ETG-4000 system. Data from each channel was first band-pass filtered between 0.01 and 0.5 Hz to remove lower or higher frequencies associated with body movements and other dynamic tissue such as respiration, heart rate, skin blood flow, etc. (Fig 3). To remove motion artifacts, we used one of the most robust methods [75], the wavelet method [74, 76] (Fig 3A). This method assumes that the obtained signal is a linear combination of the desired signal and undesired artifacts. By applying a 1-D discrete wavelet transform to each channel, details of the signal are estimated as approximation coefficients. The wavelet coefficients are assumed to have a Gaussian distribution, outliers in the distribution correspond to the coefficients related to motion artifacts, and such coefficients are set to zero. Lastly, the inverse discrete wavelet transform is applied, and the signal is reconstructed. Next, we applied the General Linear Model (GLM) implemented in HOMER-2 (Fig 3A). GLM estimates the hemodynamic response function using Gaussian basis functions and a 3rd order polynomial drift regression [74]. To correct the baseline drifts, the linear trend between the pre-trial baseline and the post-trial baseline was calculated and subtracted from values in the stimulation period as implemented in Hitachi Potato [73] (Fig 3B). Average $HbO_2$ and HHb values were obtained for the stimulation period of each trial. The range of $HbO_2$ data was significantly greater than HHb

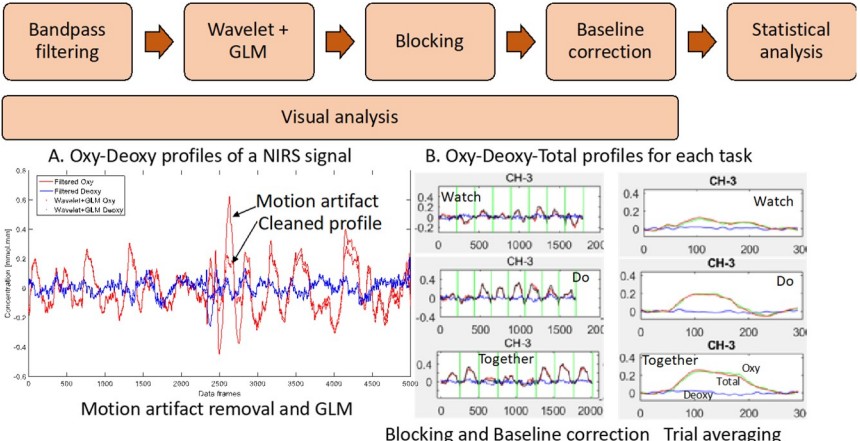

**Fig 3.** Data processing workflow (A) Filter, wavelet and GLM of NIRS signal and (B) Trial-by-trial view and average view of Oxy Hb (HbO$_2$), Deoxy Hb (HHb), and Total Hb (HbT) profiles for a given channel. (W, D, T) from 5 secs. before to 24 secs. after start of stimulation. Data have been averaged across trials and participants.

data (Fig 3B). Moreover, HbO$_2$ profiles have a greater signal to noise ratio compared to HHb and therefore, consistent with fNIRS literature, we have reported HbO$_2$ profiles [76]. In the supplementary materials section, we have also provided a visual representation of the second-to-second HbO$_2$ profile for each group (S1 Fig: TD children, S2 Fig: Children with ASD), each condition, and each channel for the entire period (pre-baseline, stimulation, and post-baseline). The pink vertical line denotes the start of the stimulation period and the data shown to the right of the pink line are the 240 frames across stimulation (10–13 s) and post-stimulation baseline (14–11 s) periods. At each step, we plotted and saved the data. Later, we visually screened the data to exclude channels with no data (flat lines) because of poor probe contact or noisy data that did not follow a canonical response typically reflective of neural activity (i.e., positive oxy and neutral to negative deoxy signal) in spite of applying the wavelet method of motion artifact removal. We also assessed whether any individual child averages for each ROI were outliers compared to the group average (< or > than 3SD) then those individual data were excluded. One child with ASD was excluded based on this criterion, hence, 14 children remained in the ASD group.

## Video data coding

A trained and reliable student researcher that was blinded to the grouping scored the session videos in order to exclude trials with significant errors. The trials were excluded from data analysis if the children did not follow task instructions or moved the cap or spoke to their partners or parents during the stimulation period. Interpersonal synchrony and motor performance were scored on a three-point scale. Spatial synchrony scores were rated from 1 to 3 with 1 = Picked incorrect blocks more than once, 2 = Picked the incorrect block once only, and 3 = Picked all blocks correctly. Temporal synchrony scores were rated from 1 to 3 with 1 = more than one block delay, 2 = One-block delay and 3 = perfect synchrony. Motor errors were counted as any of the following behaviors (i.e., two-hand use, picked more than one block at the same time, block slipping when picking or placing) with motor scores rated from 1 to 3 with 1 = more than 4 errors, 2 = 2–4 errors, 3 = 0–1 error. Two trained coders coded 20% of the video and established high levels of inter-rater reliability for all variables. Using intra-class correlations, inter-rater reliability for spatial accuracy was 86%, for temporal accuracy was 88%, and for motor accuracy was 81%. After the reliabilities were established, the primary coder coded the remaining videos for all participants.

## Data exclusion

In the TD group, 4.3% of the data were excluded due to poor fNIRS signals based on afore-mentioned criteria (i.e. dead or noisy channels during Watch = 4.4%, Do = 4.2%, and Together = 4.5%), and 6.4% of the data were excluded based on video coding (Watch = 6.4%, Do = 8.8%, and Together = 4%). For the children with ASD, 3.2% of the data were excluded due to poor fNIRS signals (Watch = 3.0%, Do = 3.6%, Together = 3.0%), and 6.29% of the data were excluded due to video coding (Watch = 7.1%, Do = 7%, Together = 4.8%). In total, 10.2% of the data were excluded in the TD group (Watch = 9.9%, Do = 12.5%, Together = 8.1%), whereas 9.3% of the data were excluded in the ASD group (Watch = 9.7%, Do = 10.6%, Together = 7.6%). There were no significant differences between the data excluded in the TD and ASD groups (all $p$s > 0.05).

## Statistical analyses

To avoid multiple channel-specific comparisons, we averaged data across channels within the same ROI based on our spatial registration output (Fig 2C and 2D show the 6 ROIs and constituent channels). We determined levels of activation for six regions of interest (ROIs) including the left and right MIFG, MSTG, and IPL regions (S1 Table shows the channel assignments for the TD and ASD groups). Using IBM SPSS, we conducted repeated-measures ANOVA using within-group factors of condition (Watch, Do, Together), hemisphere (left, right), and region of interest/ROI (MIFG, MSTG, IPL) and a between-group factor of group (TD vs ASD) for average $HbO_2$ values (SPSS, Inc.). We also used age, sex, and full-scale IQ as covariates within our analysis. Greenhouse-Geisser corrections were applied when our data violated the sphericity assumption based on Mauchly's test of sphericity. We have also conducted a channel-specific regional ANOVA that revealed similar results as the averaged channel ANOVA described above (see the S2 Table under supplementary materials for the ANOVA and the post-hoc t-test findings).

Paired t-tests were used to examine group differences in behavioral data including temporal/spatial synchrony scores, and motor scores. We also applied the False Discovery Rate (FDR) method proposed by Singh and Dan (2006) to adjust for multiple post-hoc comparisons of multichannel fNIRS data [77]. We specifically used the Benjamin-Hochberg method wherein unadjusted $p$-values are rank ordered from low to high. Statistical significance is declared if the unadjusted $p$-value is less than $p$-value threshold. $p$-value thresholds were determined by multiplying 0.05 with the ratio of the unadjusted $p$-value rank to the total # of comparisons ($p$-threshold for $i^{th}$ comparison = 0.05 x i/n; where n = total # of comparisons). In addition, we used Pearson's correlations to study associations between cortical activation and ASD severity and adaptive functioning and Spearman's rank correlations to study associations between interpersonal synchrony scores and ASD severity and adaptive functioning. FDR corrections were also used to control for multiple comparisons during correlation analyses.

## Results

### Analysis I (TD vs ASD): Quality of interpersonal synchrony and motor performance

Children with ASD had lower spatial and temporal synchrony scores compared to the TD children ($p$s < 0.01). Motor scores did not differ between groups ($p$ > 0.05). The detailed information is presented in Table 2.

**Table 2. The behavioral quality of interpersonal synchrony in TD and ASD groups.**

| Interpersonal synchrony quality | ASD (Mean ± SE) | TD (Mean ± SE) | *p*- value |
|---|---|---|---|
| Spatial synchrony | 1.94 ± 0.18 | 2.67 ± 0.08 | *p* < 0.001* |
| Temporal synchrony | 2.11 ± 0.19 | 2.74 ± 0.06 | *p* < 0.01* |
| Motor score | 2.96 ± 0.02 | 2.97 ± 0.01 | *p* > 0.05 |
| Do condition | 2.92 ± 0.03 | 2.96 ± 0.02 | *p* > 0.05 |
| Together condition | 3.00 ± 0.00 | 2.99 ± 0.01 | *p* > 0.05 |

*indicates significant differences between the ASD and TD groups. A higher value indicates better performance.

## Analysis I: Cortical activation data

The group x condition x hemisphere x region four-way repeated ANOVA revealed a significant main effect of region (F(1.9, 368.4) = 10.1, $p < 0.001$), a two-way interaction of group x region (F(1.9, 368.4) = 24.4, $p < 0.001$), a three-way interaction of group x hemisphere x region (F(2.0, 382.0) = 7.0, $p = 0.001$), and a four-way interaction of group x condition x hemisphere x region (F(3.7, 704.3) = 7.1, $p < 0.001$). The four-way interaction did not covary with age, sex or IQ (Table 3 shows the means and standard errors of HbO$_2$ concentration; and Table 4 shows the significant $p$-values and direction of effects for the post-hoc comparisons). The visual representation of averaged HbO$_2$ concentration during Watch, Do, and Together conditions in children with and without ASD is shown in Fig 4.

**Group differences.** Children with ASD had lower MIFG and MSTG but greater IPL activation compared to the TD children, and the differences are more obvious in the movement-related conditions (Do and Together) than the Watch condition (Fig 5). Specifically, during the Watch condition, children with ASD had lower right MSTG activation and greater right IPL activation than the TD children ($ps < 0.05$). During the Do condition, children with ASD had lower right MIFG, lower bilateral MSTG, and greater bilateral IPL activation compared to

**Table 3. Mean and standard error (SE) of activation based on HbO$_2$ concentration values.**

| Group activation data | Watch | | Do | | Together | |
|---|---|---|---|---|---|---|
| | **Mean** | **SE** | **Mean** | **SE** | **Mean** | **SE** |
| **TD** | | | | | | |
| *Left hemisphere* | | | | | | |
| MIFG | 0.007 | 0.004 | 0.052 | 0.004 | 0.053 | 0.005 |
| MSTG | 0.020 | 0.006 | 0.055 | 0.006 | 0.052 | 0.007 |
| IPL | -0.006 | 0.004 | 0.006 | 0.005 | 0.007 | 0.005 |
| *Right hemisphere* | | | | | | |
| MIFG | 0.011 | 0.005 | 0.041 | 0.005 | 0.053 | 0.006 |
| MSTG | 0.032 | 0.007 | 0.030 | 0.006 | 0.040 | 0.007 |
| IPL | -0.008 | 0.004 | -0.003 | 0.004 | 0.002 | 0.005 |
| **ASD** | | | | | | |
| *Left hemisphere* | | | | | | |
| MIFG | 0.006 | 0.005 | 0.038 | 0.007 | 0.050 | 0.007 |
| MSTG | 0.020 | 0.007 | 0.022 | 0.007 | 0.012 | 0.007 |
| IPL | 0.000 | 0.006 | 0.026 | 0.006 | 0.033 | 0.006 |
| *Right hemisphere* | | | | | | |
| MIFG | 0.009 | 0.005 | 0.022 | 0.006 | 0.014 | 0.006 |
| MSTG | 0.013 | 0.006 | 0.013 | 0.006 | 0.029 | 0.008 |
| IPL | 0.008 | 0.006 | 0.022 | 0.008 | 0.028 | 0.007 |

**Table 4. The significant *p*-values and direction of effects for post-hoc comparisons.**

| Comparison | Significant *p* values | Direction of effect |
|---|---|---|
| **Main effects** | | |
| Condition | < 0.001 | T > W [a] |
| | < 0.001 | D > W [a] |
| Hemisphere | 0.008 | Left > Right [a] |
| Region | < 0.001 | MIFG > IPL [a] |
| | < 0.001 | MSTG > IPL [a] |
| **Group differences** | | |
| Watch, Right MSTG | 0.018 | TD > ASD [a] |
| Watch, Right IPL | 0.011 | ASD > TD [a] |
| Do, Left MSTG | < 0.001 | TD > ASD [a] |
| Do, Left IPL | 0.005 | ASD > TD [a] |
| Do, Right MIFG | 0.007 | TD > ASD [a] |
| Do, Right MSTG | 0.023 | TD > ASD [a] |
| Do, Right IPL | < 0.001 | ASD > TD [a] |
| Together, Left MSTG | < 0.001 | TD > ASD [a] |
| Together, Left IPL | < 0.001 | ASD > TD [a] |
| Together, Right MIFG | < 0.001 | TD > ASD [a] |
| Together, Right IPL | < 0.001 | ASD > TD [a] |
| **Conditional differences** | | |
| TD, Left MIFG | < 0.001 | D > W [a] |
| | < 0.001 | T > W [a] |
| TD, Left MSTG | < 0.001 | D > W [a] |
| | < 0.001 | T > W [a] |
| TD, Left IPL | 0.035 | T > W [b] |
| TD, Right MIFG | < 0.001 | T > W [a] |
| | < 0.001 | D > W [a] |
| | 0.006 | T > D [b] |
| ASD, Left MIFG | < 0.001 | D > W [a] |
| | < 0.001 | T > W [a] |
| | 0.044 | T > D [b] |
| ASD, Left IPL | < 0.001 | D > W [a] |
| | < 0.001 | T > W [a] |
| ASD, Right MIFG | 0.040 | D > W [b] |
| ASD, Right MSTG | 0.009 | T > W [a] |
| | 0.025 | T > D [b] |
| ASD, Right IPL | 0.003 | T > W [a] |
| **Hemispheric differences** | | |
| TD, Do, MSTG | 0.001 | L > R [a] |
| ASD, Do, MIFG | 0.003 | L > R [a] |
| ASD, Together, MIFG | < 0.001 | L > R [a] |
| ASD, Together, MSTG | 0.003 | R > L [a] |

[a] indicates *p*-value < 0.05 and the effect survived FDR correction while

[b] indicates *p*-value < 0.05 but the effect did not survive FDR corrections.

the TD children (*p*s < 0.05). Similarly, during the Together condition, children with ASD had lower right MIFG, lower left MSTG and greater bilateral IPL activation compared to the TD children (*p*s < 0.001) (Fig 5).

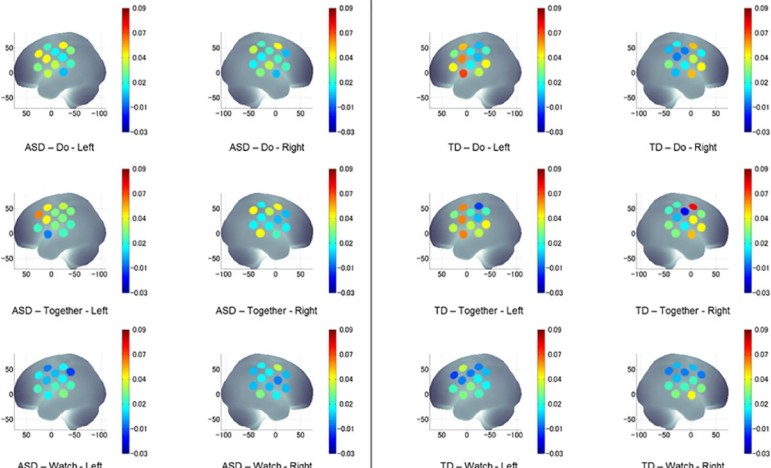

**Fig 4. A visual representation of averaged HbO₂ concentration during Watch, Do, and Together conditions in children with ASD (left) and TD children (right).** HbO₂ values on Y-axis range from 0 indicated by blue to 0.09 indicated by red and shades in between.

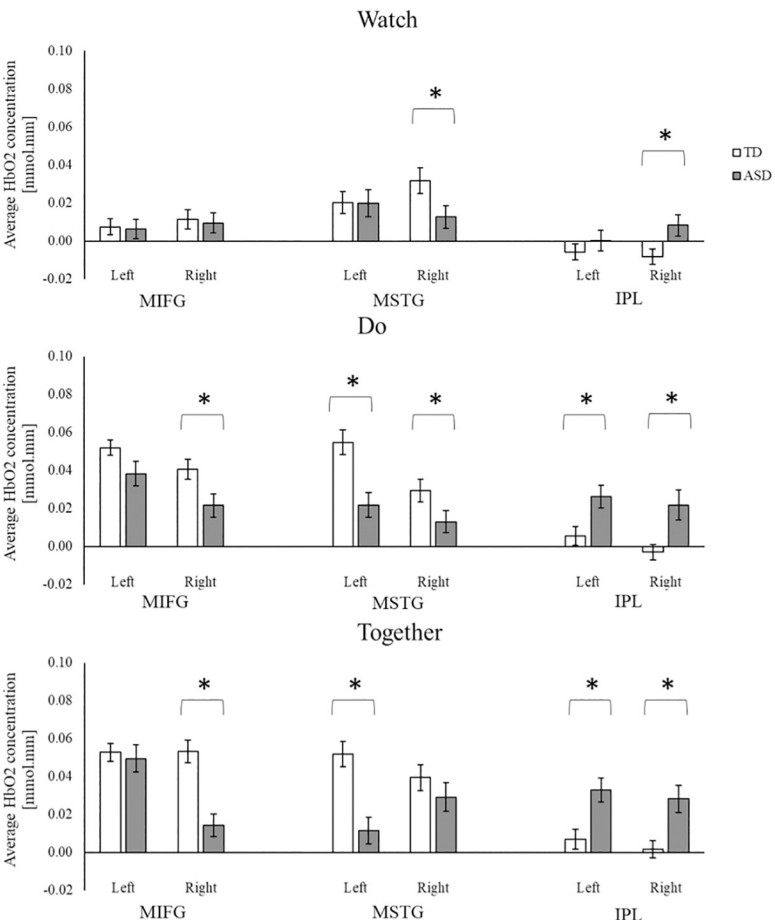

**Fig 5. Group differences in HbO₂ concentration during watch, do, and together conditions.** *indicates significant differences (i.e., $p < 0.05$ and survived for FDR correction) between the ASD and TD groups.

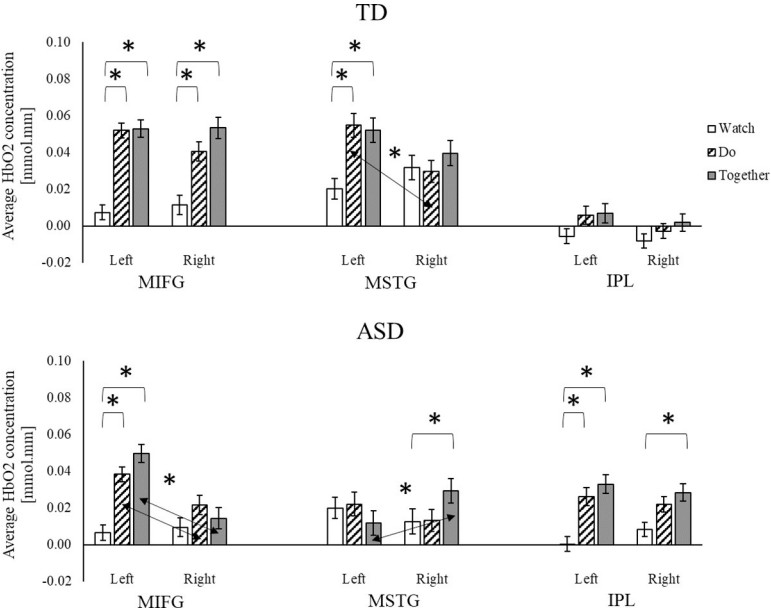

**Fig 6. Conditional and hemispheric differences in HbO₂ concentration for TD children, and children with ASD.** *
indicates conditional differences with $p < 0.05$ and surviving FDR corrections. ↔ and * indicate significant
hemispheric differences with $p < 0.05$ and surviving FDR corrections.

**Conditional differences.** Children with and without ASD showed greater activation during
movement (Do and Together) conditions compared to the observation/Watch condition (Fig 6).
Specifically, during the Together and Do compared to the Watch condition, TD children showed
greater bilateral MIFG and left MSTG activation, while the children with ASD showed greater
left MIFG and left IPL activation ($ps < 0.001$, Fig 6). We also found some trends for greater acti-
vation in the interpersonal synchrony (Together) condition compared to the solo movement
(Do) condition in both children with and without ASD, however, those comparisons did not sur-
vive FDR corrections ($ps < 0.05$, Fig 6). During the Together compared to the Do condition, the
TD children showed greater right MIFG activation while the children with ASD showed greater
left MIFG and right MSTG activation ($ps < 0.05$, Fig 6). Lastly, during the Together condition
compared to the Watch condition, the children with ASD showed greater activation in the right
MSTG and right IPL regions ($p < 0.01$. Fig 6). No other conditional differences between Watch
and Do or Do and Together were noted in the children with ASD (Fig 6).

**Hemispheric differences.** For both groups, there were no hemispheric differences for the
Watch condition (Fig 6). During the Do condition, TD children showed left-lateralized
(left > right) MSTG activation ($p = 0.001$, Fig 6) and a trend for left-lateralized MIFG activa-
tion ($p = 0.08$). Along the same lines, children with ASD showed left-lateralized MIFG activa-
tion only ($ps < 0.01$, Fig 6). During the Together condition, TD children showed similar levels
of activation in all six regions (left/right MIFG, MSTG, and IPL). However, children with ASD
showed left-lateralized MIFG activation ($p < 0.001$) and right-lateralized MSTG activation
($p = 0.003$) with no hemispheric differences in IPL activation (Fig 6).

## Analysis II correlations

**Correlations between ADOS scores, interpersonal synchrony behaviors, and cortical
activation.** The correlation between ADOS scores, interpersonal synchrony behaviors, and
cortical activation are presented in Table 5. Children with greater ADOS RRB scores (i.e.,

**Table 5. The correlations between ADOS scores, interpersonal synchrony behaviors, and cortical activation in children with ASD.**

| r- values | ADOS-SA | | | ADOS-RRB | | | ADOS-Total | | |
|---|---|---|---|---|---|---|---|---|---|
| | **W** | **D** | **T** | **W** | **D** | **T** | **W** | **D** | **T** |
| **Interpersonal synchrony Behaviors** | | | | | | | | | |
| Spatial Temporal Motor | N/A N/A N/A | N/A N/A 0.07 | -0.24* -0.19 N/A | N/A N/A N/A | N/A N/A -0.21 | −0.42** −0.43** N/A | N/A N/A N/A | N/A N/A 0.03 | −0.34** -0.26* N/A |
| **Cortical Activation** | | | | | | | | | |
| *Left hemisphere* | | | | | | | | | |
| MIFG MSTG IPL | -0.08 0.07–0.03 | −0.31** 0.21–0.13 | -0.27* -0.00–0.05 | 0.15 0.45** -0.03 | -0.10 0.38** -0.23 | 0.04 0.25* 0.17 | -0.01 0.20–0.03 | -0.28* 0.29* -0.18 | -0.20 0.08 0.02 |
| *Right hemisphere* | | | | | | | | | |
| MIFG MSTG IPL | 0.14 -0.04 -0.16 | -0.07–0.07 -0.07 | -0.09–0.21 -0.19 | 0.04 0.20 -0.18 | -0.04 0.09 -0.08 | -0.01 0.33** -0.18 | 0.13 0.04 -0.19 | -0.07–0.02–0.08 | -0.07–0.05–0.21 |

r values are presented in this figure.

* indicates $p < 0.05$

** indicates $p < 0.01$. Shaded font indicates *p*-values survived for FDR corrections. SA = social affect; RRB = repetitive behaviors.

more repetitive behaviors) also had lower spatial and temporal synchrony scores during the Together condition (r = -0.42 & -0.43, *p*s < 0.01). Similarly, children with greater ASD severity based on ADOS total scores had lower spatial synchrony scores during the Together condition (r = -0.34, *p* < 0.001). Correlations between the ADOS scores and cortical activation showed that children with ASD with greater social affective impairment had lower left MIFG activation during the Do condition (r = -0.31, *p* < 0.001). Children with ASD with more repetitive behaviors had greater left MSTG activation during the Watch and Do conditions (r = 0.45 & 0.38, *p*s < 0.001) as well as greater right MSTG activation during the Together condition (r = 0.33, *p* < 0.001). No correlations between ADOS total scores and cortical activation survived FDR corrections.

**Correlations between VABS scores, interpersonal synchrony behaviors, and cortical activation.** The correlations between VABS scores, interpersonal synchrony behaviors, and cortical activation in children with ASD are presented in Table 6. There were no correlations between VABS scores and interpersonal synchrony behaviors in children with ASD that survived FDR corrections. During the Watch condition, higher VABS communication scores were associated with greater left MSTG activation (r = 0.30, *p* < 0.001). During the Do condition, higher VABS communication scores were associated with greater left MIFG activation (r = 0.38, *p* < 0.001). During the Together condition, higher VABS communication scores were associated with greater left MIFG and right MSTG activation (rs = 0.50 & 0.48, *p*s < 0.001).

## Discussion

Children with ASD have difficulties performing socially embedded movements such as imitation [8] and interpersonal synchrony [10, 11]; which impact their relationships with peers and caregivers [13, 14] and may adversely affect their long-term social-cognitive development [13]. Despite the social significance of imitation/interpersonal synchrony skills, there is a dearth of behavioral and neuroimaging literature on neural mechanisms underlying these impairments in children with ASD. Majority of the fMRI studies reporting atypical cortical activation during imitation/interpersonal synchrony involve tasks limited to simple hand motions due to the space constraints of the MRI scanner [15, 55, 56]. Moreover, most of the aforementioned studies are limited to children and adults with low ASD severity/fewer behavioral issues because of

**Table 6. The correlations between VABS scores, interpersonal synchrony behaviors, and cortical activation in children with ASD.**

| r- values | VABS-Communication | | | VABS-Daily living | | | VABS-Socialization | | |
|---|---|---|---|---|---|---|---|---|---|
| | **W** | **D** | **T** | **W** | **D** | **T** | **W** | **D** | **T** |
| **Interpersonal synchrony Behaviors** | | | | | | | | | |
| Spatial Temporal Motor | N/A N/A N/A | N/A N/A -0.22* | -0.09 -0.10 N/A | N/A N/A N/A | N/A N/A -0.05 | 0.20 0.18 N/A | N/A N/A N/A | N/A N/A -0.19 | 0.08 0.07 N/A |
| **Cortical Activation** | | | | | | | | | |
| *Left hemisphere* | | | | | | | | | |
| MIFG MSTG IPL | 0.10 **0.30∗∗** -0.15 | **0.38∗∗** 0.17 0.07 | **0.50∗∗** 0.24* 0.14 | 0.11 0.08 0.07 | 0.07 -0.03 0.12 | 0.05 0.03 0.04 | 0.16 0.19 0.07 | 0.20 0.04 0.12 | 0.16 0.11 -0.02 |
| *Right hemisphere* | | | | | | | | | |
| MIFG MSTG IPL | -0.05 0.17 -0.05 | 0.01 0.10 -0.13 | 0.07 **0.48∗∗** 0.02 | -0.06 0.03 0.08 | -0.07 0.05 -0.02 | -0.04 0.17 -0.04 | -0.06 0.10 0.07 | -0.09 0.06 -0.16 | -0.10 0.25* -0.16 |

r values are presented in this figure.

* indicates $p < 0.05$

** indicates $p < 0.01$. Shaded font indicates $p$ values survived for FDR corrections.

the high behavioral demands of lying still within the MRI scanner. Consequently, there are more challenges for participants to comply during fMRI tasks, making it difficult for researchers to involve children with lower IQ and greater ASD severity. In fact, 22 out of 23 studies within a recent fMRI meta-analysis [78] included children with ASD with much higher IQ scores compared to the present study (mean IQ ranged from 90.7 to 116.0, present study: mean IQ = 79.57±25.35, see Table 1). Moreover, the ADOS scores of some of the fMRI studies are lower /less severe than the present study (Dougherty et al., 2016: mean ADOS total score = 11.7, SD = 3.5; Dona et al., 2017: mean ADOS comparison score = 6.5, SD = 2.2; present study: mean total score = 18.17, SD = 1.86; ADOS comparison score: 8.38, SD = 6.69, Table 1) [79, 80]. Apart from including children with wide ranging IQ and ASD severity, fNIRS has better tolerance to movement artifacts and has been used across various naturalistic movements such as walking [81] and upright arm movements [52, 61]. Using fNIRS, the present study compared the cortical activation of children with and without ASD during a naturalistic, face-to-face, reaching based interpersonal synchrony task and identified important neurobiomarkers of interpersonal synchrony impairments in children with ASD. Lastly, we correlated interpersonal synchrony behaviors and associated cortical activation with ASD severity and adaptive functioning in the children with ASD.

In line with the past studies [10, 11], children with ASD had lower spatial and temporal interpersonal synchrony accuracies compared to TD children. Moreover, their interpersonal synchrony impairments were associated with their ASD severity (those who had greater ASD severity had greater impairments in interpersonal synchrony). In terms of cortical activation (Analysis I, group differences), during interpersonal synchrony and/or its component behaviors (observation and execution), children with ASD showed reduced activation over the MIFG and MSTG regions but increased activation over the IPL regions compared to the TD children. In terms of conditional differences, children with and without ASD scaled up the cortical activation from Watch to Do and the Together condition. In terms of hemispheric differences, both TD children and children with ASD showed left-lateralized cortical activation during action execution (Do condition). However, during the Together condition, TD

children showed bilaterally symmetrical activation whereas children with ASD showed left-lateralized activation over the MIFG regions and right-lateralized activation over the MSTG regions. The correlation analyses for children with ASD (Analysis II) suggested that those who had greater ASD severity also had lower spatial and temporal accuracies during interpersonal synchrony. Greater social affect impairment in children with ASD using the ADOS was associated with lower activation over the left MIFG during the Do condition while more repetitive behaviors were associated with greater MSTG activation in all three conditions. Moreover, better communication performance in children with ASD using the VABS was associated with greater MIFG and MSTG activation during all three conditions in children with ASD.

## Similar motor coordination performance but lower interpersonal synchrony in children with ASD

We had purposely chosen a reach and clean up task as most children develop reaching abilities at a very young age, making this task universally possible across all ages. For this reason, it was not surprising that we found similar motor performance between children with and without ASD. Using motion capture systems, previous studies have shown differences in arm movement quality such as overshooting and unsmooth trajectories suggesting poor anticipatory and reactive control of skills such as reaching, catching, etc. [82, 83]. We did not utilize a motion capture system, hence, our video coding was perhaps not sensitive enough to capture the differences in visuomotor performance between children with and without ASD.

In terms of interpersonal synchrony quality, children with ASD showed reduced spatial and temporal accuracies compared to TD children and interpersonal synchrony difficulties were greater in children with higher ASD severity. Our results are consistent with previous behavioral studies of interpersonal synchrony involving marching and clapping (i.e., synchronizing march and clap actions with a partner), and pendulum swaying (i.e., swaying the pendulum in synchrony with a partner), etc. [10, 11]. Children with ASD had increased movement variability and reduced interpersonal synchrony during both marching-clapping and pendulum swaying tasks [10, 11]. Poor basic visuo-motor coordination might result in increased movement variability, which in turn makes it difficult to synchronize actions with another partner [11, 54]. Additionally, lower interpersonal synchrony could be due to poor/atypical social monitoring [4], visuo-motor [83], visual-perceptual [84], or impaired executive functions such as mental rotations, planning, inhibition, and working memory [85, 86].

## Hypoactivation of MIFG and MSTG and hyperactivation of IPL seen in children with ASD

During interpersonal synchrony and/or its component behaviors (action observation and execution), children with ASD showed lower MIFG and MSTG activation as well as greater IPL activation compared to TD children, as is often reported in the literature [15, 16, 87]. These findings fit with the literature in that children with ASD are known to have atypical activation in multiple brain regions important for imitation/interpersonal synchrony [88–90]. A rigorous meta-analysis of fMRI studies during imitation tasks have reported reduced activation in individuals with ASD in the social and object-related STS and IFG regions as well as increased activation in the motor planning-related regions of IPL [56]. Specifically, adults with ASD had hypoactivation over the frontal (i.e., inferior and middle frontal, precentral gyrus), and temporal cortices (i.e., inferior and middle temporal gyrus) [16, 89] along with hyperactivation in the inferior parietal cortices (i.e., anterior portion of the inferior parietal cortex includes the angular and supramarginal gyri) compared to those without ASD [56, 91]. During imitation of emotional expressions, unlike TD children who showed more bilateral IFG activation, children

with ASD showed no increase in fMRI-based activation in the IFG as well as hyperactivation in the left parietal and visual association cortices [90]. Our study extends these past imitation-based findings of atypical activation over the frontal, temporal, and parietal regions to inter-personal synchrony tasks in children with ASD.

The MIFG ROI in the current study captured the MFG and IFG regions. The channel specific comparisons also showed atypical activation over channels isolating the MFG and IFG ROIs in children with ASD (see S2 Table). The reach-clean up task in this study involved monitoring of partner's actions, or planning of one's own actions, or a combination of the two to clean up blocks into a container and required some level of executive functioning. The MFG region plays an important role in executive functions such as planning, response inhibition, and working memory [92]. TD children and adults show greater fMRI-based activation over the MFG during multiple executive functioning tasks, including the n-back, Go-no-go and Stroop tasks [93, 94]. In contrast, children with ASD perform poorly across multiple executive functioning tasks [85] and these difficulties have been linked to atypical activation over the prefrontal cortex [95]. On the other hand, during object-related actions, IFG plays an important role in goal understanding/mentalizing of actions during action execution and imitation [38, 40, 90, 96]. While observing object manipulations of another person, the IFG was more active when the participants focused on the goal of the motor task, and not the actions associated with the task [97]. Children with ASD are said to have difficulties understanding goals or intentions underlying their own as well as other's actions [98, 99]. The difficulties of goal understanding in children with ASD have been linked to lower IFG activation during observation of actions on objects [100]. Our findings align with other neuroimaging studies, reporting IFG hypoactivation in children with ASD during imitation tasks [15, 16]. As mentioned earlier, the reach—clean up task required some level of executive functioning to perceive, select, and retain the moment-to-moment action information from the partner and to plan one's own actions. Moreover, it required the participants to perceive relevant action information from partners and understand individual goals or shared goals of synchronizing with the tester. Hence, it is possible that the hypoactivation over the IFG and MFG regions reflects difficulties in perceiving the salience of action information from partners (i.e., due to reduced peripheral or upstream inputs), goal understanding, as well as executive functioning in children with ASD.

The MSTG ROI includes the MTG and STG regions with STS separating the two gyri. It has been suggested that STS play a role in processing the visual description of the observed action and in comparing the observed action with planned actions or what one might call visuo-motor correspondence [35, 36]. Children with ASD have lower STS activation during observation of biological motions [89] and facial emotions [101], as well as action imitation [16]. Although the current fNIRS study does not have the resolution to distinguish STS from other temporal regions (MTG and STG), it is possible that the lower MSTG activation reflects the children with ASD's difficulties in matching the observed actions with their own movement repertoire.

Besides the reduced activation over MSTG and MIFG regions, children with ASD also showed greater IPL activation during the reach-clean up task compared to TD children. Channel specific comparisons also showed increased IPL activation in children with ASD (see S2 Table). The IPL regions (including the SMG, AG, and the intra-parietal sulcus) are said to be active when planning the kinematic aspects of actions [30]. Neuroimaging studies in individuals with ASD had reported increased activation in the parietal regions during action execution, action imitation, and language-based tasks compared to TD controls [56, 102, 103]. Individuals with ASD might show greater IPL to meet the motor planning demands of the task [102]. Further research is needed to replicate the finding of hyperactivation over the IPL region in

children with ASD. Taken together, children with ASD showed atypical cortical activation (hypo/hyper activation) which fits with their behavioral impairments during the reach-clean up task. Lastly, the group differences were greater in magnitude during the action execution/ Do and interpersonal synchrony/Together conditions compared to observation/Watch condition, suggesting that the interpersonal synchrony difficulties in children with ASD mainly arise from the motor control components of interpersonal synchrony.

## Both groups increased cortical activation during interpersonal synchrony compared to its component behaviors

Both children with and without ASD showed greater activation in conditions involving action execution (Do and Together) compared to the action observation (Watch). Using fMRI, researchers have found greater activation during action imitation or execution compared to observation [40, 42, 44]. These findings echo the results of our past fNIRS study of interpersonal synchrony in healthy adults and children [22, 23]. Together, these results suggest that the challenges of imitation or interpersonal synchrony stem from the complexity of motor control components and not the observation component. We also found a trend of greater activation during interpersonal synchrony compared to its component behaviors (Watch and Do conditions) in both children with and without ASD. Similar findings were reported in previous fMRI studies with greater activation found during imitation than execution of simple finger movements [41], object manipulation [104], pantomimed actions on objects [36, 42], and communicative gestures [42]. Together, these results suggest that the cortical involvement during imitation/interpersonal synchrony is somewhat higher than pure action execution.

## Atypical lateralization during interpersonal synchrony in children with ASD

The reach and clean up task in the present study required children to use their right hand only, therefore, it is not surprising to find left-lateralized activation (contralateral to the moving limb) during action execution (Do condition) in both TD children and children with ASD [105]. Despite the right-handed nature of the reach-clean up task, the TD children showed bilaterally symmetrical activation over all ROIs during the interpersonal synchrony condition. The findings observed in TD children fit with what has been reported in the literature on how right hemisphere is more involved in action observation, visuospatial, and social information processing [35, 106, 107]. A greater involvement of the right hemisphere might lead to bilaterally symmetrical activation during imitation/interpersonal synchrony [27]. Compared to the TD children, children with ASD showed left lateralization during the solo movement condition but a different lateralization pattern during the interpersonal synchrony condition. While TD children should bilaterally symmetrical activation, children with ASD showed left-lateralized MIFG and right-lateralized MSTG activation. The atypical lateralization in children with ASD suggests that children with ASD recruited different neural circuits during interpersonal synchrony but relatively similar circuits during the solo reach-clean up task.

Children with ASD are known to have abnormal connectivity within and across cortical regions including hyper-connectivity within the frontal, parietal, and temporal cortices as well as reduced long-range connectivity between the cortices (i.e., fronto-parietal and fronto-temporal networks) [108–110]. Moreover, they also showed decreased inter-hemispheric connectivity that might alter their ability to activate both hemispheres simultaneously [111]. In fact, during an imitation task, children with ASD had decreased functional connectivity within but increased functional connectivity outside the imitation network suggesting that different networks were recruited during imitation in this population [112]. This might explain the altered

hemispheric lateralization patterns observed in children with ASD compared to the TD children.

## Associations between ASD severity and adaptive functioning and IPS behaviors/cortical activation

The behavioral performance and neural activation patterns of children with ASD are highly variable [113]. We studied how ASD severity and adaptive functioning may relate to interpersonal synchrony and associated cortical activation in children with ASD. Our findings showed that children with greater ASD severity have poor interpersonal synchrony. Children with ASD with greater social affective impairments had lower activation over the left MIFG during action execution while the presence of more repetitive behaviors was associated with greater MSTG activation during interpersonal synchrony tasks. Moreover, children with ASD with better communication performance also showed greater MIFG and/or MSTG activation during interpersonal synchrony. In short, our results suggest that the atypical activation over MIFG and MSTS regions may be linked to children's ASD severity and communicative functions.

## Limitations and future directions

The current pilot study had a relatively small sample size. We are presently adding to our study sample and hope to address the design issues of the reported study. Our study did not involve the use of a motion analysis system, and therefore, might not be sensitive enough to capture the motor impairments of the participating children with ASD. In our ongoing study, we have implemented the use of a motion tracking system to better understand the reaching impairments of children with ASD and how it might affect their interpersonal synchrony behaviors. In addition., we used 24 channels to reduce the weight of the fNIRS probes placed on the children's heads, however, this did not cover the whole brain and limited our analysis to the frontal, temporal, and parietal regions. We have now moved to using a full array of 52 channels to cover more brain regions, including sensori-motor, prefrontal, and parietal regions. This will increase our ability to isolate ROIs to specific channels. In this study, children were asked to pick up the blocks in a random order during the Do condition, which might have lower attention and arousal demands compared to the Watch and Together conditions, during which the block clean-up order was specified. To ensure a fair comparison between conditions in the ongoing study, we specify the block clean up sequence using picture cards for the Do condition.

## Clinical implications

The current study identified potential neurobiomarkers for children with ASD during interpersonal synchrony. Specifically, we found MIFG and MSTG hypoactivation as well as IPL hyperactivation in children with ASD compared to those without ASD. Moreover, MIFG and MSTG cortical activation was associated with ASD severity. The atypical activation over frontal, temporal, and parietal cortices might reflect the children with ASD's difficulties in goal understanding, executive functioning, establishing visuo-motor correspondence, as well as movement planning. Our findings suggest that ASD interventions must emphasize broader task goals and tasks requiring greater visuo-motor correspondence to improve motor and social performance. Moreover, we found greater activation differences between children with and without ASD during movement-related conditions (solo and synchronous actions), whereas during observation, both groups performed similarly. These findings suggest that sedentary interventions that involve observation and limited movements are not challenging for

children with ASD and might not lead to the greatest positive behavioral and neural change. A pilot randomized controlled trial (RCT) comparing the value of 8 weeks of rhythmic, whole-body imitation/synchrony activities to a standard of care, sedentary play intervention found that children with ASD in the rhythm group had more positive affect, spontaneous social verbalization, and greater imitation/synchrony performance following training compared to the children with ASD in the sedentary play group [114, 115]. Similarly, a more recent study using a creative yoga intervention focused on pose imitation including individual and partner poses improved the imitation and motor coordination performance of children with ASD in the yoga group compared to a sedentary play group [116]. Together, the current neuroimaging findings as well as the intervention-related behavioral changes suggest that interventions involving synchrony-based, whole-body coordination might produce equal or greater social effects than sedentary play interventions that are often provided as part of the standard of care for children with ASD. Lastly, the neurobiomarkers described in this study could assist in identifying subgroups who will most benefit from synchrony-based interventions and could be used as objective treatment response indicators of intervention outcomes.

## Supporting information

**S1 Fig.**
(PDF)

**S2 Fig.**
(PDF)

**S1 Table.**
(PDF)

**S2 Table.**
(PDF)

**S1 File.**
(ZIP)

## Acknowledgments

We would like to thank all the children and families who participated in this study. We also thank Jeffrey Eilbott for his support in training the last author in the use of Hitachi fNIRS technology during the last author's visits to the Yale Child Study Center. We also thank Dr. David Boas and his team at Boston University for training the last author on the use of Homer-2 and in sharing the MATLAB codes/functions with our group through the Boston fNIRS Training Workshop. We would also like to thank undergraduate students, Michael Hoffman, Susanna Trost, and Jessica Gibbons from the University of Delaware for their help with data collection, behavioral coding, and data analysis.

## Author Contributions

**Conceptualization:** Daisuke Tsuzuki, Kevin Pelphrey, Anjana Bhat.

**Data curation:** Anjana Bhat.

**Formal analysis:** Wan-Chun Su, McKenzie Culotta, Anjana Bhat.

**Funding acquisition:** Kevin Pelphrey, Anjana Bhat.

**Investigation:** McKenzie Culotta, Jessica Mueller, Anjana Bhat.

**Methodology:** McKenzie Culotta, Daisuke Tsuzuki, Anjana Bhat.

**Project administration:** McKenzie Culotta, Anjana Bhat.

**Resources:** Daisuke Tsuzuki, Kevin Pelphrey.

**Software:** Anjana Bhat.

**Supervision:** Anjana Bhat.

**Visualization:** Anjana Bhat.

**Writing – original draft:** Wan-Chun Su, Daisuke Tsuzuki, Anjana Bhat.

**Writing – review & editing:** Wan-Chun Su, Kevin Pelphrey, Anjana Bhat.

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
