## [Decision Letter · Decision Letter 0]

16 Apr 2020

PONE-D-19-34634

Differences in cortical activation patterns during action observation, action execution, and interpersonal synchrony between children with or without autism spectrum disorder (ASD): A functional near-infrared spectroscopy (fNIRS) study

PLOS ONE

Dear Dr. Bhat,

Thank you for submitting your manuscript to PLOS ONE. I have received reviews from three experts in the field, as well as reviewed the manuscript myself. After careful consideration, we feel that it has merit but does not fully meet PLOS ONE’s publication criteria as it currently stands.

In particular, you will see from the reviews that there are several issues that will need to be responded to before this manuscript can be considered for publication. While I will let each reviewer's comments stand on their own, I would like to highlight several concerns. First, there were several questions about your method and analytic strategy. Please increase clarity regarding the method and respond to the concerns about the contrasts and analyses used. Second, there was a concern about the logic used to make inferences from your findings. Please be sure to address this issue to ensure that your conclusion can be reasonably derived from your method and data. Third, there was a concern regarding the subgroup analysis you performed between low and high functioning ASD. While autism severity is critical to consider, you will need to address the problem of reduced power as the result of sub-setting. Please consider if sub-dividing the sample is warranted given the small cell sizes, or if autism severity can be considered in an alternative way. For example, the calibrated severity score from the ADOS could serve as a covariate, or analyzed in some other fashion. Related to this, consider whether this study should be framed as a pilot study. Finally, please be sure to carefully limit the claims made by the manuscript, especially about what these findings mean for neural mechanisms and clinical implications for ASD. Given the small sample, and that this is a relatively novel paradigm, it is important to be very cautious in the conclusions and claims.

Therefore, we invite you to submit a revised version of the manuscript that addresses the points raised during the review process. We would appreciate receiving your revised manuscript by May 31 2020 11:59PM. To enhance the reproducibility of your results, we recommend that if applicable you deposit your laboratory protocols in protocols.io, where a protocol can be assigned its own identifier (DOI) such that it can be cited independently in the future. For instructions see: http://journals.plos.org/plosone/s/submission-guidelines#loc-laboratory-protocols

We look forward to receiving your revised manuscript.

Kind regards,

Eric J. Moody, Ph.D.

Academic Editor

PLOS ONE

Journal Requirements:

1. Thank you for including your ethics statement:

'University of Delaware IRB 930721-12 Written consent obtained'

Please amend your current ethics statement to confirm that your named institutional review board or ethics committee specifically approved this study.

3. We note that Figures 1 & 2 includes an image of a [patient / participant / in the study]. 

Reviewers' comments:

Reviewer's Responses to Questions

**Comments to the Author**

1. Is the manuscript technically sound, and do the data support the conclusions?

Reviewer #1: No

Reviewer #2: Yes

Reviewer #3: Yes

2. Has the statistical analysis been performed appropriately and rigorously? 

Reviewer #1: No

Reviewer #2: Yes

Reviewer #3: Yes

3. Have the authors made all data underlying the findings in their manuscript fully available?

Reviewer #1: Yes

Reviewer #2: Yes

Reviewer #3: Yes

4. Is the manuscript presented in an intelligible fashion and written in standard English?

Reviewer #1: Yes

Reviewer #2: Yes

Reviewer #3: Yes

5. Review Comments to the Author

Reviewer #1: In this manuscript, the authors use functional near-infrared spectroscopy (fNIRS) to compare brain activation in typically developing (TD) children and children with autism spectrum disorder (ASD) during observation (“Watch”), execution (“Do”), and interpersonal synchronization (“Together”) of a block clean-up task. Brain activity was measured from three regions of interest (ROIs), including portions of superior temporal sulcus (STS), inferior frontal gyrus (IFG), and inferior parietal lobule (IPL). Behaviorally, participants with ASD displayed greater rates of both temporal and spatial errors during the Together condition, relative to TD controls. At the neural level, TD and ASD participants differed in numerous ways, including the degree of lateralization of IFG and STS activity during the Together condition. The authors also performed what appears to be an exploratory analysis dividing the ASD sample by severity, concluding that low ASD severity was associated with greater compensatory activity in IPL and less widespread hypoactivation of IFG and STS. Based on these results, the authors conclude by suggesting that their findings may provide potential biomarkers for severity of interpersonal synchrony deficits in ASD.

From a methodological perspective, the block clean-up task developed here is commendable for its real-world, naturalistic design, as well as the range of movements afforded to the participants. Unlike the highly simplified paradigms and constrained actions used in many previous studies, this task has the advantage of being highly salient and naturalistic to the participants, endowing it with greater ecological validity. Unfortunately, aside from these aspects of the task, the current study is marked by numerous issues with conceptualization, methodology, analysis, and interpretation. These concerns raise questions about the novelty and relevance of the results to the basic understanding of ASD, and the usefulness of the authors’ proposed biomarkers for identifying interpersonal synchrony deficits in ASD.

1. My first impression was that the authors planned to isolate neural correlates of interpersonal synchrony and compare them in ASD and TD: e.g. “While we know more about the neural mechanisms of imitation and how they may be affected in children with ASD; we do not know similar mechanisms for IPS impairments” (p. 2, lines 23-24). However, conceptually it is not clear to me how the Together condition reflects synchrony rather than imitation, since participants are always performing the exact same action as the tester. Could the authors please clarify how this condition isolates interpersonal synchrony per se? Also, please explain the theoretical relevance of the temporal vs. spatial measures of interpersonal synchrony.

2. Assuming that the task does measure interpersonal synchrony, the obvious comparison of interest would be the cognitive subtraction of Together – (Watch + Do). Because Watch and Do share perceptual and/or motoric elements with Together but lack the interpersonal aspect, this subtraction should better enable the identification of those processes that are specific to interpersonal synchronization.

3. Another consideration is that additional psychological factors may come into play when one is performing actions directed by another (e.g., attention, arousal). Since the current Do condition merely instructs participants to clean up the blocks “in a sequence of their choice,” it lacks this element. Therefore, to argue that the brain activity for the Together condition reflects interpersonal synchrony per se, perhaps a better comparison would be one in which the participant is still directed as to which block to pick up next (e.g., by a visual or auditory cue) but does not need to match another’s movements.

4. Why didn’t the authors collect an independent measure of motor coordination? Otherwise, it seems difficult to determine the extent to which interpersonal synchrony errors arise from impairments in social cognition vs. motor difficulties (which are known to be associated with ASD).

5. To explain the reported activations, the authors rely heavily on reverse inference from the regions being studied: for example, inferring increased reliance on one’s own motor plans in ASD based on increased IPL activation. However, this type of inference is not deductively valid (Poldrack, 2006), and is only supported to the extent that the brain region in question is selectively activated by the specified cognitive process. Unfortunately, the current study is not well-controlled either in terms of isolating specific cognitive processes or in the known selectivity of the chosen ROIs. The three brain regions of interest (IFG, STS, and IPL) are extensive, functionally heterogeneous regions, associated not only with imitation and social cognition but also with disparate cognitive processes including attention, language, and multimodal integration. Under these circumstances, the authors must exercise more caution when drawing connections between brain activity and cognitive function, especially given that the relatively low spatial resolution of fNIRS relative to fMRI. Broad statements about brain activity in participants with ASD—for example linking the reported activations in IFG to poor executive function (p. 25-26, lines 543-562) and IPL to motor planning (p. 26-27, lines 571-585)—are unwarranted and should be removed unless they can be supported with specific behavioral evidence from the participants themselves.

6. The analysis of ASD severity feels completely post-hoc and should be removed. The sample is already extremely small and underpowered, and little or no theoretical justification is provided for dividing it into two smaller groups. The interpretation of the statistical analysis is also questionable. If there is no significant difference in behavioral interpersonal synchrony scores between the LASD and HASD groups, it is simply incorrect to state that “the TD group had the best IPS performance, followed by the LASD group and lastly, the HASD group” (p. 21, lines 455-456). With respect to cortical differences between groups, the authors lean heavily on the idea of compensatory activity. However, this inference seems largely speculative, and should either be further qualified or removed from reporting of the results.

7. Video data coding: Why was only a single coder used? Can the authors explain why they chose what seems like a fairly coarse 3-point scale? What were the types of additional movements coded during the stimulation period, and why is this measure of interest?

8. Statistical analyses: please include specific values of statistical tests and p values. Were paired t-tests corrected for multiple comparisons, and, if so, how? Finally, please note that figures should NOT indicate statistical significance for comparisons that fail to survive FDR correction.

9. Particularly given the neuroanatomical context, the decision to use the abbreviation “IPS” for interpersonal synchrony made the manuscript much more confusing (IPS = intraparietal sulcus). Please remove this abbreviation from the manuscript.

Reviewer #2: This study examined interpersonal synchrony and cortical activation during naturalistic reach and clean up tasks in children with and without ASD. The fNIRS experimental design, spatial registration and statistical modeling are strengths. There are several weaknesses which would need to be addressed before the manuscript can be considered further for publication. I enumerate specific comments below.

Major comments

The N (14 ASD, 17 TD) seems low to me for a study of this sort (NIRS with ASD). Perhaps the study should be framed as a pilot study?

Line 56 - Please consider including a citation as an example of this body of literature.

The hypotheses are not clear (starting on Line 174). First, the authors refer to conditions, e.g. ‘Together’ and ‘Do’ which have not been previously defined. Second, activation patterns for one group are described with no explanation about mention of the other group (hemispheric differences). Finally, the severity hypothesis refers to impairments which have not been defined (do the authors expect impairments for children with ASD across the board? And if so what are they?).

Some details of participant inclusion/exclusion are not clear.

How was ASD diagnosis ruled out in the TD children? Were SCQ scores taken into consideration?

Were children with ASD and preterm birth included?

Data exclusions for motion and other artifacts are described in multiple places. It would be good to see an aggregate of all excluded data, per group and a between-group statistical comparison. It is also not clear how a ‘significant motion artifact’ was defined in the video coding process.

The term ‘IFG’ (inferior frontal gyrus) is used to refer to the frontal ROI. This ROI includes channels in middle frontal gyrus and the pre/post central gyrus. Using the term IFG to refer to this ROI seems misleading as it is not representative of the underlying anatomy. The authors should consider a more representative term.

I agree with the authors that considering autism severity is very important. However,, I do not understand the rationale behind the subgrouping analysis. There are several flaws to the approach taken which was to divide the larger autism group based on a cut point of ADOS severity scores (low vs high). First, the dividing line between high and low functioning is somewhat arbitrary. Second, the two groups are small and therefore statistical analyses are underpowered. The authors could have instead examined correlations between autism severity and outcome variables. On a related note, the comparison of ASOS scores between the two groups is redundant since the groups were defined based on their ADOS scores. I won’t comment on the severity findings/interpretation because I am not convinced on the validity of the subgroup definitions.

Did the sample of children with ASD included in the present study actually represent lower functioning children than the samples included in most fMRI studies? I agree that an important benefit of NIRS is its ability to include those who can not undergo MRI scanning and potentially that population includes lower functioning children with autism. Knowing where the sample in the present study fits will be important for interpreting the results.

The conclusion that starts on line 540 seems like an overgeneralization and it is also not clear what the authors mean by ‘cortical atypicalities.’

The paragraph starting on line 543 describes neuro-functioning related to executive function in adults, then describes differences in children with autism. The authors should consider citing executive function studies in TD children as a reference point.

Minor comments

The authors refer to gender on Line 187 but I think they mean sex.

Which form of the Vineland was used (survey or caregiver interview)?

Use of ‘IPS’ for interpersonal synchrony might be confusing in the context of this paper because it is also commonly used for intraparietal sulcus. Since the abbreviation of brain regions (IPL, STS, etc) are commonplace in neuroimaging and already used in the current paper I suggest changing IPS. Perhaps using the term synchrony (after appropriately defining the term) would be better since it is the only type of synchrony investigated here.

Reviewer #3: Re: PONE-D-19-34634

Bhat et al have examined the fNIRS activation patterns associated with interpersonal synchrony in ASD. This is a novel approach, using a technique allowing monitoring of brain activity during more naturalistic behavior that is possible with techniques such as fMRI, particularly useful in examining this aspect of behavior, so it is of interest. I do have a few comments, though.

First, in the abstract, for ‘In terms of group differences in cortical activation’ and for ‘Subgroup analysis revealed that children with high ASD severity had a more widespread activation…’- would add ‘during IPS’ just for clarity.

Introduction- the argument is proposed that the motor aspect might have primacy, culminating in ‘children with ASD might have impaired social monitoring and poor planning/incoordination that could affect their ability to imitate…’- would be VERY cautious about the motor aspect. Children with developmental conditions affecting coordination in isolation do not have ASD-like behavior in this regard. Certainly, ASD does have significant motor findings- but it seems more appropriate to discuss the motor component in the context of a more circumspect question as to its role. Also, briefly explain the ‘pendulum swaying tasks’ so that the reader knows how it is an IPS task. Same thing for the ‘finger tapping task’. For the studies cited late in paragraph 6, might point out which of these are EEG- as earlier in the paragraph ‘fMRI studies have reported’ is stated, but ‘increased theta activity’ presumably follows a transition to EEG studies. Later ‘there are few studies utilizing fMRI in children with ASD’- actually there are a growing number of such studies these days, with improvements in ways to habituate to the environment. The fNIRS is fairly unique for its role for this particular task, though. Finally, ‘For the hemispheric differences, the TD children would have bilateral activation during IPS/Together condition’- did the authors intend to contrast this with ASD children?

Methods- could somewhere the be a demographics table for high vs low ASD as with Table 1? Maybe even as part of Table 1? It seems that the ‘fNIRS cap embedded with two 3 x 3 probe sets’ is better demonstrated in Figure 2A, rather than the stated Figure 1A. Please expand for clarity so the reader understands what is done beyond ‘For the Together condition, the tester led the block clean up in a random order while the participant followed by picking up the same block as the tester’- the participants were specifically asked to follow along with the tester? How was this instructed? Also, not sure as to the role of the phrase ‘To be clear’. How does ‘Two 3x3 probe sets, consisting of five infrared emitters and for receivers’ result in 24 channels? Please clarify. Finally, it seems remarkable that more data was eliminated from the TD group than the ASD group- deserves brief comment regarding the Visual data coding.

Results- somewhere, would include the full statistics for what is presented in Table 2 (in text, or in the table). Same with the text regarding Fig 5 and 6. Finally, in Fig 6, it seems that the ASD STS Together L vs R should also have an asterisk for that comparison.

Discussion- for ‘limited to children and adults with low ASD severity because of the high behavioral demands of lying still’ – would also add challenges with complying with tasks. Later, again, ‘marching in clapping’ task- how that is an IPS task?- in addition to the pendulum swaying mentioned above in the Introduction comments. Also, ‘The increased movement variability stems from poor visuo-motor coordination that makes it difficult to synchronized actions with another partner’- see above in the Introduction comments for cautionary note on this presumption of motor primacy. Top of page 25, ‘During IPS and/or its component behaviors…’- might clarify which component behaviors are being addressed here, as the point of this paper is that the data is scant for IPS itself. Not sure the need for the ‘;’ after ‘have reported reduced activation in individuals with ASD’. Top of page 26, should be pointed out that the executive function argument for IFG hypoactivation might be rendered moot if the decreased salience due to difficulties understanding the shared goals predominates, so there is no understanding of a need to allocate executive function resources. Later in page 26 ‘difficulties processing observed motions’- might be rendered moot due to the same salience issue. Middle of page 27 end of paragraph, would change ‘from the motor components of IPS’ to ‘from the motor regulations components of IPS’. End of page 29, maybe ‘poor visuo-motor correspondence’ should, it seems, reflect the interpersonal/social aspect, as their own internal visuo-motor correspondence might be fine. Same issue with the same text at the top of page 31. Finally, for the RCT, for the ‘whole-body coordination activities’- are these imitative tasks, as is stated for the yoga intervention?

6. PLOS authors have the option to publish the peer review history of their article (what does this mean?). If published, this will include your full peer review and any attached files.

Reviewer #1: No

Reviewer #2: No

Reviewer #3: Yes: David Q. Beversdorf, MD

---

## [Author Response · Author response to Decision Letter 0]

22 Jul 2020

We sincerely appreciate the extensive feedback from the reviewers in improving the quality of this manuscript. We have modified the manuscript based on reviewer comments and as a result has further strengthened the manuscript! Each edit within the manuscript is highlighted using track changes. Changes within the manuscript are listed here by providing the page # and line #. 

Reviewer 1

Comment 1: My first impression was that the authors planned to isolate neural correlates of interpersonal synchrony and compare them in ASD and TD: e.g. “While we know more about the neural mechanisms of imitation and how they may be affected in children with ASD; we do not know similar mechanisms for IPS impairments” (p. 2, lines 23-24). However, conceptually it is not clear to me how the Together condition reflects synchrony rather than imitation, since participants are always performing the exact same action as the tester. Could the authors please clarify how this condition isolates interpersonal synchrony per se? Also, please explain the theoretical relevance of the temporal vs. spatial measures of interpersonal synchrony.

Response: We have reworded the sentence the reviewer commented on to as follows: “Previous fMRI studies investigated cortical activation in children with ASD during finger/hand movement imitation; however, we do not know whether these findings generalize to naturalistic face-to-face imitation/interpersonal synchrony” (page 2, lines 26-29).

We appreciate the chance to differentiate imitation from interpersonal synchrony. When we refer to action imitation we are implying copying of discrete actions, whereas interpersonal synchrony of actions implies sustained synchronous movements that coincide with a partner’s actions; for example, drumming motions of two adults as they engage in a drum circle activity. During imitation, the form or spatial accuracy of the copied movement is analyzed, whereas interpersonal synchrony of actions involves synchronous movement over time and allows us to analyze both the movement form/spatial accuracy (i.e., are both individuals moving with the same form/range of movement) and temporal accuracy (i.e., are both individuals moving similarly over time or with a similar speed). Theoretically, the processes involved in imitating and engaging in interpersonal synchrony of actions should be similar (i.e., observing the partner, reproducing the partner’s actions). However, the challenge of synchronizing is greater than that of imitation due to the continuous nature of action-related interpersonal synchrony. We believe that the Together condition in the present study resembles interpersonal synchrony (and not imitation) due to the continuous nature of the reach and cleanup task. 

We have added more details describing the similarities and differences between imitation and interpersonal synchrony in the introduction section (pages 3-4, lines 63-77). Lastly, we have tried to explain the theoretical relevance of scoring both spatial and temporal accuracy during interpersonal synchrony tasks (see intro section, page 4, lines 77-83). 

Given the reviewer comments, it appears that their definition of interpersonal synchrony (IPS) is different from ours in that they are considering the sense of connectedness as interpersonal synchrony. However, IPS for us is the behavior of sustained matching of continuous actions.

Comment 2: Assuming that the task does measure interpersonal synchrony, the obvious comparison of interest would be the cognitive subtraction of Together – (Watch + Do). Because Watch and Do share perceptual and/or motoric elements with Together but lack the interpersonal aspect, this subtraction should better enable the identification of those processes that are specific to interpersonal synchronization.

Response: As we mentioned in the introduction section, Watch and Do are component behaviors of the Together condition. For both imitation and interpersonal synchrony, one needs to perceive the cues from the environment and partner (or Watch), and then anticipate and reactively adjust one’s own actions to the environment (or Do) (see Semin and Cacioppo, 2009; Vesper et al., 2010). This is also suggested in the condition-related differences (W v D v T), we have reported in our previous papers (Bhat et al., 2017; Su et al., 2020) and current paper: During observation/Watch, we mainly see greater localized superior temporal activation during the Watch condition whereas in the Do condition there is significant multi-region activation, and in the Together condition we usually see slightly greater multi-region and bilateral activation. Overall, the challenge of together condition is more similar to Do (action execution) than the Watch (action observation). 

Using the dynamical systems perspective of motor control/development, interpersonal synchrony or imitation behaviors in children result from interactions between perceptual-social, motor, and cognitive systems. We agree that the whole is not a mere sum of the parts. We believe in embodied accounts of imitation/synchrony and value the contributions of all three perceptual-social, motor, and cognitive system and don’t think that interpersonal synchrony is simply the added cognitive challenge of the Together condition compared to Watch only + Do only. Interpersonal synchrony is a complex behavior involving multiple subsystems and should be considered as a whole along with its social-cognitive and perceptuo-motor challenges. 

Comment 3: Another consideration is that additional psychological factors may come into play when one is performing actions directed by another (e.g., attention, arousal). Since the current Do condition merely instructs participants to clean up the blocks “in a sequence of their choice,” it lacks this element. Therefore, to argue that the brain activity for the Together condition reflects interpersonal synchrony per se, perhaps a better comparison would be one in which the participant is still directed as to which block to pick up next (e.g., by a visual or auditory cue) but does not need to match another’s movements.

Response: We agree that the instructions in the Do condition were not ideal as the children were free to choose the cleanup sequence with fewer attentional demands compared to the Watch and Together conditions (when they were required to observe the adult partner). We have acknowledged this limitation in the discussion section under “Limitations” (page 34, lines 723-727). In fact, we have another ongoing study where we are re-doing this task with an improved task design. In the new study (which is currently halted due to the COVID crisis), the Do condition, involves the child seeing a specific order of blocks on a picture card and the child is asked to cleanup according to the order shown (page 34, lines 723-727). 

Comment 4: Why didn’t the authors collect an independent measure of motor coordination? Otherwise, it seems difficult to determine the extent to which interpersonal synchrony errors arise from impairments in social cognition vs. motor difficulties (which are known to be associated with ASD).

Response: In the current study, we have behaviorally coded the motor accuracy during the Together/interpersonal synchrony and the Do/motor execution conditions. A motor error was identified when the child dropped a block or knocked over the container. We found that temporal and spatial accuracies during the Together condition differed between children with and without ASD whereas motor error/accuracy was similar between the two groups. This is not surprising because the reaching task is an everyday task that children with ASD are able to perform successfully from a very young age. In the new study, we have incorporated more sensitive measures of motion analysis (using inertial measurement units) to obtain accurate measures of motor coordination/performance. The limitation specified by the reviewer is now stated within the Limitations section (page 34, lines 714-719).

Comment 5: To explain the reported activations, the authors rely heavily on reverse inference from the regions being studied: for example, inferring increased reliance on one’s own motor plans in ASD based on increased IPL activation. However, this type of inference is not deductively valid (Poldrack, 2006), and is only supported to the extent that the brain region in question is selectively activated by the specified cognitive process. Unfortunately, the current study is not well-controlled either in terms of isolating specific cognitive processes or in the known selectivity of the chosen ROIs. The three brain regions of interest (IFG, STS, and IPL) are extensive, functionally heterogeneous regions, associated not only with imitation and social cognition but also with disparate cognitive processes including attention, language, and multimodal integration. Under these circumstances, the authors must exercise more caution when drawing connections between brain activity and cognitive function, especially given that the relatively low spatial resolution of fNIRS relative to fMRI. Broad statements about brain activity in participants with ASD—for example, linking the reported activations in IFG to poor executive function (p. 25-26, lines 543-562) and IPL to motor planning (p. 26-27, lines 571-585)— are unwarranted and should be removed unless they can be supported with specific behavioral evidence from the participants themselves.

Response: We agree that given the lower spatial resolution of fNIRS and the study design, we cannot be certain about the roles of the different regions of interest. We have reworded the section referenced by the reviewer and avoid assigning functions to certain cortical regions.

To address reviewer concerns, we have also conducted channel-specific regional comparisons because certain channels were able to isolate specific ROIs. For example, 99.4% of the centroid formed by channel 3 is over the MFG region, therefore, we are more confident about using channel 3 as a representative channel for the MFG ROI. Overall, the results of the channel-specific regional comparisons are similar to that of the averaged-channel regional comparisons, further supporting the accuracy of our regional assignment (see more details on spatial registration and channel assignment under S1 table in Supplementary Materials; and the statistical results of channel specific analyses under S2 table in Supplementary Materials).

Comment 6: The analysis of ASD severity feels completely post-hoc and should be removed. The sample is already extremely small and underpowered, and little or no theoretical justification is provided for dividing it into two smaller groups. The interpretation of the statistical analysis is also questionable. If there is no significant difference in behavioral interpersonal synchrony scores between the LASD and HASD groups, it is simply incorrect to state that “the TD group had the best IPS performance, followed by the LASD group and lastly, the HASD group” (p. 21, lines 455-456). 

Response: We agree that given the small sample size of the present study it is best to leave out the subgroup analysis in children with ASD. To address this problem and to provide more information on how ASD severity and level of adaptive functioning relate to interpersonal synchrony behaviors and cortical activation, we are now reporting correlations between ADOS/VABS scores and interpersonal synchrony behaviors and cortical activation (see Results section on pages 24-26, lines 495-526). 

Comment 7: With respect to cortical differences between groups, the authors lean heavily on the idea of compensatory activity. However, this inference seems largely speculative, and should either be further qualified or removed from reporting of the results.

Results: We have attempted to reword the aforementioned section and have toned down the statement (page 31, lines 650-651). 

Comment 8: Video data coding: Why was only a single coder used? Can the authors explain why they chose what seems like a fairly coarse 3-point scale? What were the types of additional movements coded during the stimulation period, and why is this measure of interest?

Response: We had established inter-rater reliability for all variables between two coders, but the primary coder coded all the videos to reduce scoring variability. While two coders are used to establish reliability and develop a more robust coding scheme, it is ideal for one reliable coder to code the entire dataset, if the dataset is small enough; which was the case for this study. Specifically, two coders coded 20% of the data and established reliability using intra-class correlations. Inter-rater reliability for spatial accuracy score was 85.7%, for temporal accuracy score was 88.1%, and motor accuracy score was 81%). These details are now added to the coding section of the manuscript (page 18, lines 387-390).

In terms of the coding scheme, we used a more qualitative 3-point Likert scale to quantify the spatial and temporal accuracy of interpersonal synchrony behaviors. In our new study, we are planning to code the actual number of spatial and temporal errors to increase the scoring range. In addition, we are also using motion tracking systems to obtain quantitative measures of synchronous motions between the child’s and the tester’s arm movements. We have acknowledged the qualitative measure of synchrony as a limitation and suggest modifications that we have made within our currently ongoing study (page 34, lines 714-719). 

We also agree that the additional movements are not the main focus of the present study since we have excluded trials with significant movement artifacts. To avoid confusion, we have removed the additional movement findings from the manuscript.

Comment 9: Statistical analyses: please include specific values of statistical tests and p values. Were paired t-tests corrected for multiple comparisons, and, if so, how? Finally, please note that figures should NOT indicate statistical significance for comparisons that failed to survive FDR correction.

Response: The mean and SE of HbO2 concentration as well as specific p-values are listed within the Tables 3 and 4 in the main body of the document. We used the FDR method developed by Singh and Dan (2006) to obtain significance thresholds for multiple comparisons. More specifically, we adjusted the p-thresholds by multiplying 0.05 with the ratio of the p-value rank to the total # of comparisons (p-threshold for ith comparison = 0.05 x i/n; where n = total # of comparisons). We then ranked the post-hoc p-values from the smallest to the largest and compared them with the FDR corrected thresholds (also ranked from smallest to the largest). If the p-value of a certain post-hoc t-test is less than the FDR corrected threshold, statistical significance was declared. Lastly, we have removed the differences that did not survive FDR corrections from the figures.

Comment 10: Particularly given the neuroanatomical context, the decision to use the abbreviation “IPS” for interpersonal synchrony made the manuscript much more confusing (IPS = intraparietal sulcus). Please remove this abbreviation from the manuscript.

Response: Thanks for the suggestion. We agree that the abbreviation “IPS” might be confusing. We have removed this abbreviation and use “interpersonal synchrony” throughout the manuscript. 

Reviewer 2:

Comment 1: The N (14 ASD, 17 TD) seems low to me for a study of this sort (NIRS with ASD). Perhaps the study should be framed as a pilot study?

Response: We agree to frame the present study as a pilot study. We have changed the title to, “Differences in cortical activation patterns during action observation, action execution, and interpersonal synchrony between children with or without autism spectrum disorder (ASD): An fNIRS pilot study” 

Comment 2: Line 56 - Please consider including a citation as an example of this body of literature.

Response: Two fMRI studies that focus on the arm/finger movement imitation are cited (Wadsworth et al., 2016; Jack & Morris, 2014, see page 3, line 60) to support the aforementioned statement.

Comment 3: The hypotheses are not clear (starting on Line 174). First, the authors refer to conditions, e.g. ‘Together’ and ‘Do’ which have not been previously defined. Second, activation patterns for one group are described with no explanation about mention of the other group (hemispheric differences). Finally, the severity hypothesis refers to impairments which have not been defined (do the authors expect impairments for children with ASD across the board? And if so what are they?).

Response: Thanks for pointing out deficiencies in the hypothesis paragraph. We have now clarified the conditions of Watch, Do and Together in the introduction section (page 9, lines 200-203). We have added a hypothesis on hemispheric differences in cortical activation for children with ASD (pages 9-10, lines 203-205). Based on reviewer comments, we have removed subgroup analyses and instead conduct correlations. We have added correlational hypotheses as well (page 10, lines 207-209).

Comment 4: Some details of participant inclusion/exclusion are not clear. How was ASD diagnosis ruled out in the TD children? Were SCQ scores taken into consideration? Were children with ASD and preterm birth included? 

Response: We have added our list of inclusion/exclusion criteria in the “participants” paragraph to clarify further. To rule out ASD diagnosis in the TD children, we conducted a screening interview with the parent to ensure that the children had no ASD, other neurodevelopmental diagnosis, developmental delays, significant birth history, or a family history of ASD, as well as no vision, hearing, motor, or language delays. The SCQ and ADOS was only used with the children with ASD, but we have asked both groups to complete the VABS questionnaire. The TD children in this study had normal range of adaptive functioning, and no significant behavioral issues. Preterm birth was ruled out in all TD participants (pages 10-11, lines 217-228).

Comment 5: Data exclusions for motion and other artifacts are described in multiple places. It would be good to see an aggregate of all excluded data, per group and between-group statistical comparison. It is also not clear how a ‘significant motion artifact’ was defined in the video coding process.

Response: We agree that it would be better to provide the total amount of data excluded. We excluded data after checking signal quality and after behavioral coding of video data. We are now reporting aggregated findings across these two analyses at the end of the methods session (page 18, lines 392-401). t-tests have been run for between-group comparisons of excluded data. There were no significant differences in the amount of excluded data between children with and without ASD (all ps > 0.05).

Video coding was mostly used to assess task compliance and was not used to determine exclusions due to motion artifacts. We removed motion artifacts programmatically using the wavelet method, which is the most robust method reported so far. This method assumes that the obtained signal is a linear combination of the desired signal and undesired artifacts. By applying a 1-D discrete wavelet transform to each channel, details of the signal are estimated as approximation coefficients. The wavelet coefficients are assumed to have a Gaussian distribution, outliers in the distribution correspond to the coefficients related to motion artifacts, and such coefficients are set to zero. During visual analysis of the fNIRS signal, we excluded channels with no data (flat lines) because of poor probe contact. We also excluded channels that were noisy in spite of applying the wavelet method. Noisy data were those that did not follow a canonical response typically reflective of neural activity (i.e., positive oxy and neutral to negative deoxy signal). Later, we also assessed whether any individual child averages for each ROI were outliers compared to the group average (< or > than 3SD) then those individual data were excluded. One child with ASD was excluded on this basis.

Comment 6- The term ‘IFG’ (inferior frontal gyrus) is used to refer to the frontal ROI. This ROI includes channels in middle frontal gyrus and the pre/post central gyrus. Using the term IFG to refer to this ROI seems misleading as it is not representative of the underlying anatomy. The authors should consider a more representative term.

Response: We agree that the term IFG is not representative of the frontal regions we covered. We have changed IFG to MIFG in the revised manuscript. Similarly, we have also renamed STS to MSTG as it covers the Superior and Middle Temporal Gyrus (or the STS/sulcus in between the two gyri). 

Comment 7: I agree with the authors that considering autism severity is very important. However, I do not understand the rationale behind the subgrouping analysis. There are several flaws to the approach taken which was to divide the larger autism group based on a cut point of ADOS severity scores (low vs high). First, the dividing line between high and low functioning is somewhat arbitrary. Second, the two groups are small and therefore statistical analyses are underpowered. The authors could have instead examined correlations between autism severity and outcome variables. On a related note, the comparison of ASOS scores between the two groups is redundant since the groups were defined based on their ADOS scores. I won’t comment on the severity findings/interpretation because I am not convinced on the validity of the subgroup definitions.

Response: We agree that given our small sample size, subgrouping of the ASD sample is not a good idea; hence, we have removed the subgrouping analysis. To provide more information on how ASD severity/level of functioning is associated with interpersonal synchrony behavior/cortical activation, we have correlated ADOS/VABS scores and interpersonal synchrony behaviors/cortical activation and this is now reported in the results section (pages 24-26, lines 499-528). 

Comment 8: Did the sample of children with ASD included in the present study actually represent lower functioning children than the samples included in most fMRI studies? I agree that an important benefit of NIRS is its ability to include those who cannot undergo MRI scanning and potentially that population includes lower functioning children with autism. Knowing where the sample in the present study fits will be important for interpreting the results.

Response: The current study included children with high ASD severities and low adaptive functions. More specifically, 6 out of the 14 ASD children had an ADOS comparison score of 10, indicating the very high ASD severity and a VABS total score ≤1% with low adaptive functioning. These children with ASD would certainly have difficulties participating in fMRI studies but were able to tolerate the fNIRS cap as well as the naturalistic face to face interactive tasks.

Comment 9: The conclusion that starts on line 540 seems like an overgeneralization and it is also not clear what the authors mean by ‘cortical atypicalities.’

Response: We agree, we have reworded the statement to emphasize the similarities between the findings of the current study and Yang and Hoffmann’s meta-analysis (page 29, lines 600-605).

Comment 10: The paragraph starting on line 543 describes neuro-functioning related to executive function in adults, then describes differences in children with autism. The authors should consider citing executive function studies in TD children as a reference point.

Response: We have now cited a meta-analysis paper in TD children (McKenna et al., 2017) to better support our statement on the role of prefrontal cortex for executive functioning (page 30, line 614-616).

Comment 11: The authors refer to gender on Line 187 but I think they mean sex.

Response: Yes, we intended to report the biological sex of the participants and have changed the word “gender” to “sex” throughout the manuscript.

Comment 12: Which form of the Vineland was used (survey or caregiver interview)?

Response: We have used the parent-report survey of the Vineland and this is now mentioned on pages 11-12, lines 244-250. 

Comment 13: Use of ‘IPS’ for interpersonal synchrony might be confusing in the context of this paper because it is also commonly used for intraparietal sulcus. Since the abbreviation of brain regions (IPL, STS, etc) are commonplace in neuroimaging and already used in the current paper I suggest changing IPS. Perhaps using the term synchrony (after appropriately defining the term) would be better since it is the only type of synchrony investigated here.

Response: Thanks for the suggestion. We agree that the abbreviation “IPS” might cause confusion since it is a common abbreviation for a cortical region. We had removed all the abbreviations and replaced them with the term “interpersonal synchrony” throughout the manuscript. 

Reviewer 3: 

Comment 1: First, in the abstract, for ‘In terms of group differences in cortical activation’ and for ‘Subgroup analysis revealed that children with high ASD severity had a more widespread activation…’- would add ‘during IPS’ just for clarity.

Response: Based on comments from other reviewers, we have removed the subgroup analysis and instead report correlations between ASD severity/adaptive functioning and behaviors and cortical activation during interpersonal synchrony. Hence, the abstract has been rewritten. 

Comment 2: Introduction- the argument is proposed that the motor aspect might have primacy, culminating in ‘children with ASD might have impaired social monitoring and poor planning/incoordination that could affect their ability to imitate…’- would be VERY cautious about the motor aspect. Children with developmental conditions affecting coordination in isolation do not have ASD-like behavior in this regard. Certainly, ASD does have significant motor findings - but it seems more appropriate to discuss the motor component in the context of a more circumspect question as to its role. 

Response: We have reworded the introduction to distinguish impairments in socially-embedded actions of children with ASD (page 3, lines 53-54). 

Comment 3: Introduction: Also, briefly explain the ‘pendulum swaying tasks’ so that the reader knows how it is an IPS task. Same thing for the ‘finger tapping task’. For the studies cited late in paragraph 6, might point out which of these are EEG, as earlier in the paragraph ‘fMRI studies have reported’ is stated, but ‘increased theta activity’ presumably follows a transition to EEG studies. 

Response: We are happy to clarify. We have added more details on how the pendulum swaying and finger tapping tasks are testing interpersonal synchrony. Specifically, during the pendulum swaying task, both the child and the tester were asked to sway a pendulum antero-posteriorly while synchronizing their pendulum motions (Fitzpatrick et al., 2016, page 8, lines 161-165). During the finger-tapping task, the child synchronized his/her finger tapping movements with a partner or a computer using auditory feedback (Kawasaki et al., 2017, page 8, lines 178-179). We also point out the EEG study to do a better transition from multiple fMRI studies (page 8, line 176).

Comment 4: Introduction: Later ‘there are few studies utilizing fMRI in children with ASD’- actually there are a growing number of such studies these days, with improvements in ways to habituate to the environment. The fNIRS is fairly unique for its role for this particular task, though. 

Response: We agree that there are a growing number of fMRI studies in children with ASD. We have changed the sentence to, “Furthermore, although there are a growing number of studies utilizing fMRI in the children with ASD, the fMRI testing environment is still challenging for children with ASD, leading to greater anxiety due to its loud noise and narrow space in the scanner bore.” (page 9, lines 188-190).

Comment 5: Introduction- Finally, ‘For the hemispheric differences, the TD children would have bilateral activation during IPS/Together condition’- did the authors intend to contrast this with ASD children?

Response: We have now added hypotheses on hemispheric differences in cortical activation for children with ASD (pages 9-10, lines 203-205). 

Comment 6: Methods- could somewhere the be a demographics table for high vs low ASD as with Table 1? Maybe even as part of Table 1? 

Response: Due to the small sample size and reviewer recommendations, we have removed the subgroup analysis from the paper, therefore, the demographic distribution of high vs. low ASD is not added to Table 1. Instead we ran correlations to study how ASD severity and level of functioning might affect the children with ASD’s interpersonal synchrony behaviors and associated cortical activation (see results section, pages 24-26, lines 496-527). 

Comment 7: Methods- It seems that the ‘fNIRS cap embedded with two 3 x 3 probe sets’ is better demonstrated in Figure 2A, rather than the stated Figure 1A. 

Response: Thanks for pointing this out. We have made sure to refer to the closer view presented in Figure 2A when describing the probe placements (page 14, line 295).

Comment 8: Methods- Please expand for clarity so the reader understands what is done beyond ‘For the Together condition, the tester led the block clean up in a random order while the participant followed by picking up the same block as the tester’- the participants were specifically asked to follow along with the tester? How was this instructed? Also, not sure as to the role of the phrase ‘To be clear’. 

Response: Instructions were provided before the trial started and the verbal instructions used are now clearly stated in the manuscript (page 13, lines 270-277). We have also deleted the phrase “to be clear” from the sentence.

Comment 9: Methods- How does ‘Two 3x3 probe sets, consisting of five infrared emitters and for receivers’ result in 24 channels? Please clarify. 

Response: Each probe set consists of 5 emitters and 4 receivers that are alternately placed in the probe holder. A channel is defined as the midpoint between each emitter and receiver pair. Therefore, there are 24 channels in total (12 on each hemisphere, see figure below). More information has been added to the manuscript to clarify how we end up with 24 channels (page 14, lines 299-301).

Comment 10: Methods- Finally, it seems remarkable that more data was eliminated from the TD group than the ASD group- deserves brief comment regarding the visual data coding.

Response: We have checked the percent of data excluded. In total, 10.17% of the data were excluded in the TD group (9.90% during Watch, 12.48% during Do, 8.13% during Together), whereas 9.32% of the data were excluded in the ASD group (9.7% during Watch, 10.62% during Do, 7.64% during Together). There were no significant differences between the data excluded in the TD and ASD groups (all ps > 0.05) (pages 18-19, lines 392-402). 

Comment 11: Results: somewhere, would include the full statistics for what is presented in Table 2 (in text, or in the table). Same with the text regarding Fig 5 and 6. Finally, in Fig 6, it seems that the ASD STS Together L vs R should also have an asterisk for that comparison.

Response: The full statistics for spatial and temporal accuracies are provided in Table 2 (page 21). For cortical activation data shown in Fig 5 and 6, the significant p-values and direction of effects for post-hoc comparisons are listed in table 4 on pages 21-22). Lastly, we have added an asterisk to Figure 6 to show the significant difference between left and right temporal activation. 

Comment 12: Discussion: for ‘limited to children and adults with low ASD severity because of the high behavioral demands of lying still’ – would also add challenges with complying with tasks.

Response: We agree and have added challenges of task compliance as a limitation as well (page 9, lines 188-190). 

Comment 13: Discussion- Later, again, ‘marching in clapping’ task- how that is an IPS task?- in addition to the pendulum swaying mentioned above in the Introduction comments. 

Response: During the marching and clapping task, the children were asked to synchronize both upper and lower limb movements with that of the tester’s march-clap actions. We have added more clarifying details in the intro and discussion session where these studies are mentioned (page 29, lines 586-588).

Comment 14: Discussion: Also, ‘The increased movement variability stems from poor visuo-motor coordination that makes it difficult to synchronize actions with another partner’- see above in the Introduction comments for cautionary note on this presumption of motor primacy. 

Response: We agree that the difficulties in visuo-motor coordination might not be the only reason that lead to high movement variability and poor interpersonal synchrony skills. We have re-written this section more carefully and describe the multiple potential reasons for children with ASD’s interpersonal synchrony difficulties. (page 28, lines 585-586 and page 35, lines 733-736). 

Comment 15: Top of page 25, ‘During IPS and/or its component behaviors…’- might clarify which component behaviors are being addressed here, as the point of this paper is that the data is scant for IPS itself. 

Response: We have specified the component behaviors of IPS including action observation and motor execution (page 29, lines 597).

Comment 16: Not sure the need for the ‘;’ after ‘have reported reduced activation in individuals with ASD’. 

Response: We had deleted the “,” in the sentence (page 29, line 598).

Comment 17: Consider that the executive function argument for IFG hypoactivation might be rendered moot if the decreased salience due to difficulties understanding the shared goals predominates, so there is no understanding of a need to allocate executive function resources. Later in page 26 ‘difficulties processing observed motions’- might be rendered moot due to the same salience issue.

Response: ASD is a multisystem disorder with abnormal connectivity across multiple cortical regions (frontal, temporal, parietal, etc.). It is difficult to favor one mechanism over another when multiple ROIs are showing atypical activation in children with ASD including reduced MFG, IFG, MSTG activation, and greater IPL activation. At this time, we do not have a single unifying theory to explain why these different ROIs show hypo/hyper-activation. Without overanalyzing the results, we have tried to explain that the task involved motor planning/execution, goal-directed anticipation, and visuo-motor correspondence for successful task performance. Children with ASD showed deficiencies in regions that contribute to these aforementioned processes, albeit along with many other brain regions known to be important for imitation/synchrony. 

The MFG region plays a role in executive functioning, while the IFG region plays a role in goal understanding. We have also conducted channel-specific regional comparisons (see spatial registration paragraph on page 16, lines 333-340, and more details within the S2 table under supplementary materials). For example, 99.4% of the centroid formed by channel 3 is over the MFG region, therefore, we are more confident about using channel 3 as a representative channel for MFG. Overall, the results of the channel-specific regional comparisons are similar to that of the averaged-channel regional comparisons, further supporting the accuracy of our regional assignment. Using channel-specific comparisons, we have found that there is atypical activation over both IFG and MFG regions in children with ASD. So, at this point we are unable to choose one neurobiomarker over the other. Both ROIs, MFG and IFG are showing atypicalities. For this reason, we have written the discussion more generally and do not emphasize one theory over another. 

Comment 19: Middle of page 27 end of paragraph, would change ‘from the motor components of IPS’ to ‘from the motor regulation components of IPS’. End of page 29, maybe ‘poor visuo-motor correspondence’ should, it seems, reflect the interpersonal/social aspect, as their own internal visuo-motor correspondence might be fine. Same issue with the same text at the top of page 31.

Response: We have changed “from the motor components of interpersonal synchrony” to “from the complexity of motor control components of interpersonal synchrony” (page 32, line 657 and 666). We have reworded the term “visuo-motor correspondence” to “visuo-motor correspondence in an interpersonal context”. We do believe that visuo-motor correspondence is required for interpersonal synchrony (page 35, line 735). We also know that visuo-motor coordination is impaired in children with ASD even during solo actions (i.e., movements are slower and inaccurate) compared to TD children, mentioned on page 29, lines 590-591).

Comment 20: Finally, for the RCT, for the ‘whole-body coordination activities’- are these imitative tasks, as is stated for the yoga intervention?

Response: Yes, in the yoga intervention study, the trainer used pose imitation as well as partner poses to promote motor skills in children with ASD. In the rhythm intervention, children were imitating and synchronizing whole-body actions performed to the beat of music or songs. We have changed the term “whole-body coordination activities” to “whole-body imitation/synchrony activities” to emphasize the imitative/synchrony aspects of the rhythm intervention (see page 36, line 744). 

We hope we have addressed reviewer concerns to their satisfaction and look forward to hearing from them.

Best,

Anjana Bhat, PT, PhD

University of Delaware

---

## [Decision Letter · Decision Letter 1]

27 Aug 2020

PONE-D-19-34634R1

Differences in cortical activation patterns during action observation, action execution, and interpersonal synchrony between children with or without autism spectrum disorder (ASD): An fNIRS pilot study

PLOS ONE

Dear Dr. Bhat,

Thank you for submitting your manuscript to PLOS ONE. After careful consideration, we feel that it has merit but does not fully meet PLOS ONE’s publication criteria as it currently stands. Therefore, we invite you to submit a revised version of the manuscript that addresses the points raised during the review process.

As you will see from the Reviewers' comments, there are still a few minor points that will need clarification. Please address each of these comments, and submit your revised manuscript by Oct 11 2020 11:59PM. If you will need more time than this to complete your revisions, please reply to this message or contact the journal office at plosone@plos.org. Please include the following items when submitting your revised manuscript:

We look forward to receiving your revised manuscript.

Kind regards,

Eric J. Moody, Ph.D.

Academic Editor

PLOS ONE

Reviewers' comments:

Reviewer's Responses to Questions

**Comments to the Author**

1. If the authors have adequately addressed your comments raised in a previous round of review and you feel that this manuscript is now acceptable for publication, you may indicate that here to bypass the “Comments to the Author” section, enter your conflict of interest statement in the “Confidential to Editor” section, and submit your "Accept" recommendation.

Reviewer #1: All comments have been addressed

Reviewer #2: (No Response)

Reviewer #3: (No Response)

2. Is the manuscript technically sound, and do the data support the conclusions?

Reviewer #1: (No Response)

Reviewer #2: Yes

Reviewer #3: Yes

3. Has the statistical analysis been performed appropriately and rigorously? 

Reviewer #1: (No Response)

Reviewer #2: Yes

Reviewer #3: Yes

4. Have the authors made all data underlying the findings in their manuscript fully available?

Reviewer #1: (No Response)

Reviewer #2: Yes

Reviewer #3: Yes

5. Is the manuscript presented in an intelligible fashion and written in standard English?

Reviewer #1: (No Response)

Reviewer #2: Yes

Reviewer #3: Yes

6. Review Comments to the Author

Reviewer #1: (No Response)

Reviewer #2: The authors have done a thorough job addressing my concerns. I only have two remaining comments.

(original comment 7) I appreciate the switch from subgroup analysis to correlations but I do not see any mention of control for multiple comparisons in this section. Have the authors considered multiple comparisons in their correlation analysis?

(original comment 8) The authors have addressed my comments but I still do not have a sense of how this cohort compares to cohorts in fMRI studies of ASD. Could the authors provide a brief summary of severity across fMRI studies (or at least the ones they cite) and how that compares to the present study’s cohort? I also do not see any corresponding changes in the manuscript. I think this information is important to include for all readers of the manuscript.

Reviewer #3: Re: PONE-D-19-34634 R1

Bhat et al have examined the fNIRS activation patterns associated with interpersonal synchrony in ASD. This is a novel approach, using a technique allowing monitoring of brain activity during more naturalistic behavior that is possible with techniques such as fMRI, particularly useful in examining this aspect of behavior, so it is of interest. The authors have addressed nearly all of the comments of this reviewer. Just one minor issue could be clarified.

Discussion- The new text in lines 629-635 does a better job of accounting for the potential of perceptual and motor issues both being salient. However, it is still worth a bit of emphasis, and maybe could be covered by adding at the end of that paragraph something like ‘One cannot exclude the possibility that, due to impaired perception of the salience of the action information from the partner, that the input is decreased upstream of the IFG and MFG (from higher order perceptual inputs), contributing to hypoactivation.’ As an obvious example for demonstration of this point, someone with cortical blindness would have hypoactivation in these regions in their effort to do these tasks, and impaired performance, obviously, but not resulting from an executive functioning problem. This would also tie in nicely with the new text in the subsequent paragraph.

7. PLOS authors have the option to publish the peer review history of their article (what does this mean?). If published, this will include your full peer review and any attached files.

Reviewer #1: No

Reviewer #2: No

Reviewer #3: **Yes: **David Q. Beversdorf, MD

---

## [Author Response · Author response to Decision Letter 1]

13 Sep 2020

Thank you for all your efforts in reviewing and improving this manuscript. We were delighted to see that all three reviewers recognized our efforts in addressing their concerns from the previous round, “All comments have been addressed.”, “The authors have done a thorough job addressing my concerns”, and “The authors have addressed nearly all of the comments of this reviewer”. 

Here we address the last few reviewer concerns. We have further modified the manuscript based on their comments. Each edit within the manuscript is highlighted using track changes. Changes within the manuscript are listed here by providing the page # and line #. 

Reviewer 2, Comment 1: I appreciate the switch from subgroup analysis to correlations but I do not see any mention of control for multiple comparisons in this section. Have the authors considered multiple comparisons in their correlation analysis?

Response: Yes, we used the False Discovery Rate (FDR) method for fNIRS-based analysis proposed by Singh and Dan (2006) to account for multiple comparisons during post-hoc testing as well as correlational analyses. We specifically used the Benjamin-Hochberg method wherein unadjusted p-values are rank ordered from low to high. p-value thresholds were determined by multiplying 0.05 with the ratio of the unadjusted p-value rank to the total # of comparisons (p-threshold for ith comparison = 0.05 x i/n; where n=total # of comparisons). Statistical significance is declared if the unadjusted p-value is less than the p-value threshold. We have added this clarification descriptions within the statistical analysis section (Page 20, Line 426-427).

Comment 2: The authors have addressed my comments but I still do not have a sense of how this cohort compares to cohorts in fMRI studies of ASD. Could the authors provide a brief summary of severity across fMRI studies (or at least the ones they cite) and how that compares to the present study’s cohort? I also do not see any corresponding changes in the manuscript. I think this information is important to include for all readers of the manuscript.

Response: The current fNIRS study was able to include children with lower IQ and greater ASD severity assessed using the ADOS compared to participants in most fMRI studies. In a meta-analysis of fMRI findings in individuals with ASD (Philip et al., 2012), among the 23 studies that reported IQ in children with ASD, only 1 study included children with lower IQ (Mean IQ= 76.8, Okem et al., 2000). The rest of the studies had a sample mean IQ ranging from 90.7 to 116.0 (SD ranged from 12.3 to 27.7, Philip et al., 2012). Similarly, in a more recent meta-analysis of fMRI studies of ASD during observation and imitation tasks, the IQ scores were much higher than that of the present study (our mean IQ was 79.6 � 25.4). Specifically, the sample mean IQ ranged from 93.3 to 124.8 for the 13 studies included in the analysis. 

The ADOS scores for some of the fMRI studies were lower (indicating less severe ASD) than the present study (Dougherty et al., 2016: mean ADOS total score = 11.7, SD = 3.5; Dona et al., 2017: mean ADOS comparison score = 6.5, SD = 2.2; present study: mean ADOS total score = 18.2, SD = 1.9; ADOS comparison score: 8.4, SD = 6.7, Table 1). We have added this information comparing fMRI studies and the present study within the discussion session (Page 27, line 543-554).

Reviewer 3, Comment 1: The new text in lines 629-635 does a better job of accounting for the potential of perceptual and motor issues both being salient. However, it is still worth a bit of emphasis, and maybe could be covered by adding at the end of that paragraph something like ‘One cannot exclude the possibility that, due to impaired perception of the salience of the action information from the partner, that the input is decreased upstream of the IFG and MFG (from higher order perceptual inputs), contributing to hypoactivation.’ As an obvious example for demonstration of this point, someone with cortical blindness would have hypoactivation in these regions in their effort to do these tasks, and impaired performance, obviously, but not resulting from an executive functioning problem. This would also tie in nicely with the new text in the subsequent paragraph. 

Response: Thank you for suggesting the possibility of reduced perception of action information from partners leading to hypoactivation in the MFG and IFG regions. We have added your suggested phrasing within the discussion session (Page 31, line 648-649). 

We look forward to hearing from you and the reviewers. 

Thank you for your time and consideration!

---

## [Decision Letter · Decision Letter 2]

24 Sep 2020

Differences in cortical activation patterns during action observation, action execution, and interpersonal synchrony between children with or without autism spectrum disorder (ASD): An fNIRS pilot study

PONE-D-19-34634R2

Dear Dr. Bhat,

We’re pleased to inform you that your manuscript has been judged scientifically suitable for publication and will be formally accepted for publication once it meets all outstanding technical requirements.

Kind regards,

Eric J. Moody, Ph.D.

Academic Editor

PLOS ONE

Additional Editor Comments (optional):

Reviewers' comments:

Reviewer's Responses to Questions

**Comments to the Author**

1. If the authors have adequately addressed your comments raised in a previous round of review and you feel that this manuscript is now acceptable for publication, you may indicate that here to bypass the “Comments to the Author” section, enter your conflict of interest statement in the “Confidential to Editor” section, and submit your "Accept" recommendation.

Reviewer #3: All comments have been addressed

2. Is the manuscript technically sound, and do the data support the conclusions?

Reviewer #3: Yes

3. Has the statistical analysis been performed appropriately and rigorously? 

Reviewer #3: Yes

4. Have the authors made all data underlying the findings in their manuscript fully available?

Reviewer #3: Yes

5. Is the manuscript presented in an intelligible fashion and written in standard English?

Reviewer #3: Yes

6. Review Comments to the Author

Reviewer #3: All of my remaining comments have been addressed. Apparently I have to keep typing till I reach 100 characters.

7. PLOS authors have the option to publish the peer review history of their article (what does this mean?). If published, this will include your full peer review and any attached files.

Reviewer #3: **Yes: **David Q. Beversdorf, MD

---

## [Editor Report · Acceptance letter]

6 Oct 2020

PONE-D-19-34634R2 

Differences in cortical activation patterns during action observation, action execution, and interpersonal synchrony between children with or without autism spectrum disorder (ASD): An fNIRS pilot study 

Dear Dr. Bhat:

I'm pleased to inform you that your manuscript has been deemed suitable for publication in PLOS ONE. Congratulations! Your manuscript is now with our production department. 

Kind regards, 

on behalf of

Dr. Eric J. Moody 

Academic Editor

PLOS ONE